# USE: A Unified Self-Ensembling Framework for Test-Time Prompt Tuning

**Siru Jiang** [1 2]   **Jian Liang** [2 3]   **Ran He** [2 3]   **Tieniu Tan** [1 2 4]

## Abstract

Test-time adaptation (TTA) has emerged as a popular paradigm for improving the performance of vision–language models (*e.g.*, CLIP) on downstream tasks. Among existing CLIP-based TTA methods, Test-Time Prompt Tuning (TPT) is a pioneering work that optimizes textual prompts using multiple test-time augmentations and remains a strong baseline to date. In this work, we revisit TPT and reveal that its optimization can be interpreted as implicitly learning from self-generated pseudo labels. Building on this perspective, we propose a unified self-ensembling framework (USE) that ensures consistency between the optimization and inference stages. During optimization, we introduce a simple yet effective self-ensembling (SE) strategy that emphasizes the test image itself over its augmented views adaptively to obtain more reliable pseudo labels. To fully exploit the potential of augmentations, we further apply the same strategy at inference time, unifying the objectives of both stages. Notably, SE can also act as a lightweight optimization-free TTA method. Extensive experiments across multiple datasets demonstrate that SE and USE outperform their counterparts, respectively. Furthermore, SE yields consistent performance gains when integrated with existing TTA methods. The code is available at
https://github.com/sirujiang/USE.

## 1. Introduction

Vision-language models (VLMs) pre-trained on large-scale image-text pairs, such as CLIP (Radford et al., 2021) and ALIGN (Jia et al., 2021), have gained widespread adoption in classical vision tasks, including image classification (Zhou et al., 2022b; Khattak et al., 2023), semantic segmentation (Li et al., 2022; Xu et al., 2022), and image captioning (Hu et al., 2023; Ramos et al., 2023). Despite the better generalization ability than traditional vision models, they are still vulnerable to distribution shifts (Mayilvahanan et al., 2024) in downstream tasks. Continuous efforts have been made to adapt VLMs to downstream tasks with few labeled data while maintaining parameter efficiency (Zhou et al., 2022b; Khattak et al., 2023; Gao et al., 2024; Zanella & Ben Ayed, 2024a). As labeled data are not always available in practice, a growing line of work has resorted to test-time adaptation (TTA) (Sun et al., 2020; Liang et al., 2025), which performs model adaptation using only unlabeled test samples (Shu et al., 2022; Feng et al., 2023; Zanella & Ben Ayed, 2024b; Farina et al., 2024).

As a pioneering work, Test-Time Prompt Tuning (TPT) (Shu et al., 2022) produces multiple augmentations for a single test image and minimizes marginal entropy (Zhang et al., 2022a) over the averaged predictions to improve model performance. Due to its simplicity and effectiveness, the framework has been widely adopted in various TTA methods, including (Feng et al., 2023; Yoon et al., 2024; Sheng et al., 2025b). However, a recent benchmark (Sheng et al., 2025a) conducts a fair comparison between different TTA methods, revealing that the classic TPT remains a surprisingly strong and hard-to-beat baseline.

In this work, we thoroughly examine the optimization objective of TPT and decompose it by introducing an auxiliary distribution $q$, resulting in a reverse cross-entropy (RCE) term and a KL divergence term as follows.

$$\mathcal{L}_{mem} = \underbrace{-p\log(q)}_{\text{RCE}} - p\log\left(\frac{p}{q}\right) \quad \begin{array}{l} \rightarrow \text{avg. prediction} \\ \rightarrow \text{pseudo label} \end{array}$$

In the context of TPT, $p$ represents the average prediction of the selected low-entropy augmented views. Let $q$ be a stop-gradient version of $p$, the KL divergence with respect to $p$ has already achieved its optimum. Consequently, the marginal entropy is effectively equivalent to the remaining RCE loss. Throughout this paper, we adopt the terminology of self-training (Zou et al., 2019; Cascante-Bonilla et al.,

[1]School of Advanced Interdisciplinary Sciences, University of Chinese Academy of Sciences [2]NLPR & MAIS, Institute of Automation, Chinese Academy of Sciences [3]School of Artificial Intelligence, University of Chinese Academy of Sciences [4]Nanjing University. Correspondence to: Jian Liang <liangjian92@gmail.com>.

*Proceedings of the 43rd International Conference on Machine Learning*, Seoul, South Korea. PMLR 306, 2026. Copyright 2026 by the author(s).

2021) and refer to $q$ as a pseudo label. To this end, we obtain a simplified variant of TPT that implicitly encourages augmented samples to align with self-generated pseudo labels. Building on this perspective, a question naturally arises:

***How can we improve the quality of the pseudo label q?***

In TPT and its following literature (Yoon et al., 2024; Dafnis & Metaxas, 2025), $q$ is implicitly constructed by averaging predictions from augmented views. Through empirical analysis, we find that the original image plays a distinctive role in zero-shot image classification, yet often underweighted or even excluded. We propose a simple yet effective self-ensembling strategy (SE) that explicitly aggregates predictions from both strong and weak augmentations, thereby providing a balanced pseudo label. To further emphasize the weak augmentation (the test image itself), we devise an entropy-based weighting coefficient that adjusts the contribution of the test image adaptively.

Unlike most existing TTA methods, which rely solely on the weak augmentation for final inference, R-TPT (Sheng et al., 2025b) demonstrates that incorporating strong augmentations significantly enhances adversarial robustness. Inspired by this finding, we adopt the aforementioned self-ensembling strategy, aggregating predictions from both weak and strong views at inference time. This design ensures both stages within a unified self-ensembling framework (USE)[1], ensuring consistency between the pseudo-label estimation step and the final inference step. Notably, SE alone can be used as an optimization-free TTA method.

Extensive experiments on ImageNet datasets and fine-grained datasets demonstrate that SE and USE outperform their respective counterparts. Besides, SE is a lightweight module that integrates with existing TTA methods seamlessly, consistently enhancing their performance. Our contributions are summarized as follows:

- We reinterpret TPT from a new perspective, revealing that it implicitly learns from its self-generated pseudo label.

- We devise SE, a simple yet effective self-ensembling pseudo-label estimation strategy that could also serve as a standalone optimization-free TTA alternative.

- We propose USE, a unified TTA framework that ensures consistency between the optimization objective and the final inference objective.

- Extensive results on multiple datasets validate the effectiveness of both USE and SE. Moreover, SE provides consistent performance gains when applied to other methods without significant computational overhead.

---

[1]To avoid any ambiguity, we unify the optimization and inference stages into a single framework, and we **separately** ensure consistency between them.

## 2. Related Work

### 2.1. Vision–Language Models

Pretrained on large-scale image–text pairs, vision–language models (VLMs) (Radford et al., 2021; Jia et al., 2021; Zhai et al., 2023; Yao et al., 2022), have become increasingly popular in recent years. For example, CLIP (Radford et al., 2021) aligns visual and textual modalities within a shared embedding space, which enables it to perform image classification tasks using hand-crafted text templates and unlabeled test data in a zero-shot manner. However, despite their strong generalization, VLMs still suffer from performance degradation due to distribution shifts. Consequently, a wide range of parameter-efficient fine-tuning (PEFT) strategies have been proposed, including adapter (Zhang et al., 2022b; Gao et al., 2024), prompt tuning (Zhou et al., 2022b;a; Khattak et al., 2023), and low-rank adapter (LoRA) (Hu et al., 2022; Wang et al., 2025), enabling effective adaptation by updating a small proportion of model parameters. Although these methods are predominantly designed for few-shot adaptation, some prior work has extended their application to VLM adaptation using unlabeled data (Huang et al., 2022; Tanwisuth et al., 2023; Liang et al., 2024; Dong et al., 2025).

### 2.2. Test-Time Augmentation

Data augmentation (Yin et al., 2019; Hendrycks et al., 2020) is a widely adopted training technique that enhances generalization by applying random transformations to input data. Test-time augmentation (TTAug) (Shanmugam et al., 2021; Kim et al., 2020; Lyzhov et al., 2020) extends this technique to inference, improving model accuracy (He et al., 2016), robustness (Smith & Gal, 2018), and uncertainty (Pérez et al., 2021). A standard practice is to average predictions across multiple augmented views (He et al., 2016), while more advanced approaches further improve performance by selecting informative augmentations (Kim et al., 2020) or reweighting different views (Shanmugam et al., 2021).

With the growing popularity of VLMs, recent efforts have been devoted to developing TTAug for CLIP (Zanella & Ben Ayed, 2024b; Li et al., 2025; Song et al., 2025), exploring augmentations from both textual and visual perspectives. On the text side, CLIP naturally supports prompt ensembling (Radford et al., 2021), and subsequent works further extend this paradigm by constructing and reweighting text prompts (Allingham et al., 2023; Song et al., 2025). On the vision side, MTA (Zanella & Ben Ayed, 2024b) assigns adaptive weights to different image augmentations via mean-shift clustering, while ZERO (Farina et al., 2024) stabilizes predictions by marginalizing over multiple augmented views. Our method identifies the importance of weak augmentation and emphasizes it while explicitly reweighting strong augmentations adaptively, providing a simple yet effective alternative.

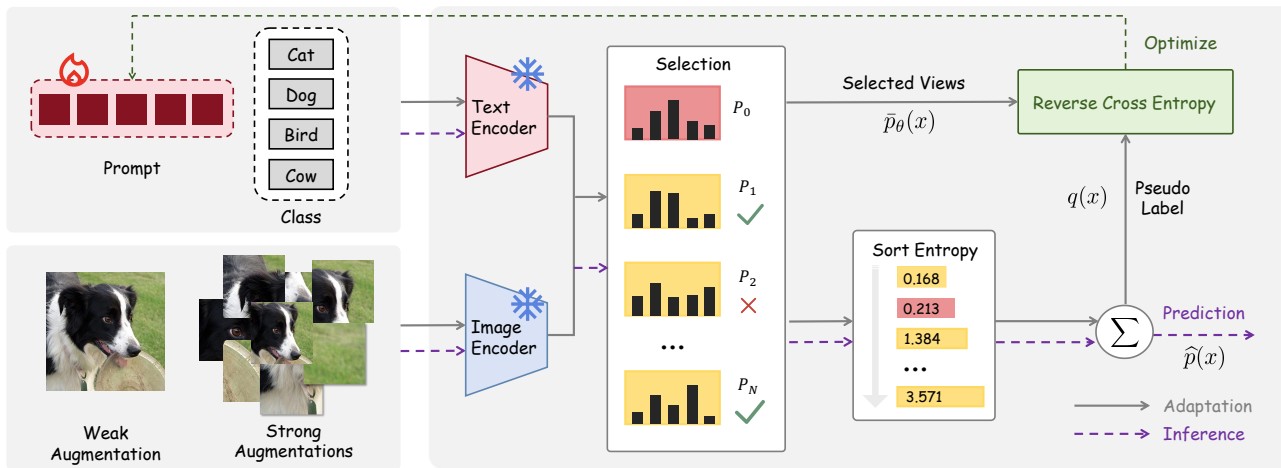

*Figure 1.* The pipeline of the proposed framework. During optimization, we use reverse cross-entropy to optimize the textual prompt. Specifically, the pseudo-label is estimated via the proposed SE strategy, which prioritizes the weak augmentation and adaptively incorporates strong augmentations. By applying the same SE strategy to generate the final prediction during inference, USE ensures consistency between the two stages. Notably, SE can also function independently as an optimization-free TTA method.

## 2.3. Test-Time Adaptation

Test-time adaptation (TTA) aims to improve model performance by leveraging unlabeled test samples during inference, without access to source data or additional retraining (Liang et al., 2025; Dong et al., 2025; Yan et al., 2026). While online TTA methods (Wang et al., 2021; Iwasawa & Matsuo, 2021), which continuously adapt the model over a stream of test data, episodic TTA methods, such as TTT (Sun et al., 2020) and MEMO (Zhang et al., 2022a), perform adaptation independently for each test instance. Research has emerged exploring TTA methods for CLIP in both scenarios (Qian & Hu, 2024; Ma et al., 2023; Karmanov et al., 2024; Shu et al., 2022; Samadh et al., 2023; Maharana et al., 2025; Zhou et al., 2025; Yu et al., 2024).

In particular, we focus on episodic TTA (Shu et al., 2022; Samadh et al., 2023) and restrict our discussion under this setting. On the one hand, TTA methods that do not require backpropagation are commonly referred to as training-free methods (Farina et al., 2024; Zanella & Ben Ayed, 2024b), which were discussed before. On the other hand, as a representative training-based work, TPT (Shu et al., 2022) optimizes the textual prompt by marginal entropy minimization, which is inspired by MEMO (Zhang et al., 2022a). Subsequent works further extend this framework (Yoon et al., 2024; Sheng et al., 2025b), for instance, C-TPT (Yoon et al., 2024) introduces an extra calibration-aware objective. Beyond textual prompt tuning, STS explores spectral subspace steering on textual embeddings (Dafnis & Metaxas, 2025), TPS (Sui et al., 2025) modulates visual class prototypes, and TTL (Imam et al., 2025) introduces low-rank adapters in the visual branch. In contrast, we revisit the classic TPT method from a self-learning perspective and propose a consistent framework that aligns the objectives of both stages.

## 3. Method

We present a Unified Self-Ensembling framework (USE) for CLIP in this paper, and its overview is illustrated in Figure 1. Section 3.1 revisits TPT (Shu et al., 2022), a classic yet hard-to-beat baseline, and re-examines its underlying mechanism from a self-learning perspective. Section 3.2 introduces the proposed SE, which prioritizes weak augmentation and ensembles it with strong augmentations adaptively. Section 3.3 presents USE, a consistent framework that bridges the objectives of the optimization and inference stages.

### 3.1. Revisit TPT from a Pseudo-Labeling Perspective

A recent study (Sheng et al., 2025a) reveals that the classic entropy-minimization-based method, Test-Time Prompt Tuning (TPT) (Shu et al., 2022), remains competitive. Therefore, we begin by briefly reviewing it, which consists of an optimization stage and an inference stage.

In the optimization stage, given a test image $x$, TPT generates $(N-1)$ randomly augmented views via AugMix (Hendrycks et al., 2020), forming a set of strong augmentations $\{\mathcal{A}_i(x)\}_{i=1}^{N-1}$, along with a weak augmentation $\mathcal{A}_0(x)$. The prediction $p_\theta(\mathcal{A}_i(x))$ is obtained by CLIP with parameters $\theta$ on the $i$-th augmented view of the input $x$. Views with Shannon entropy lower than $\tau$, which is the $\rho$-th percentile threshold, are selected as confident views. TPT computes their average as $\bar{p}_\theta(x)$ below,

$$\bar{p}_\theta(x) = \frac{1}{N_\rho} \sum_{i=0}^{N-1} \mathbb{I}[\mathbf{H}(p_\theta(\mathcal{A}_i(x))) \leq \tau] \, p_\theta(\mathcal{A}_i(x)), \quad (1)$$

where $\mathbf{H}(\cdot)$ denotes the Shannon entropy, and $N_\rho = \lfloor \rho N \rfloor$ is the number of selected augmented views. To encourage

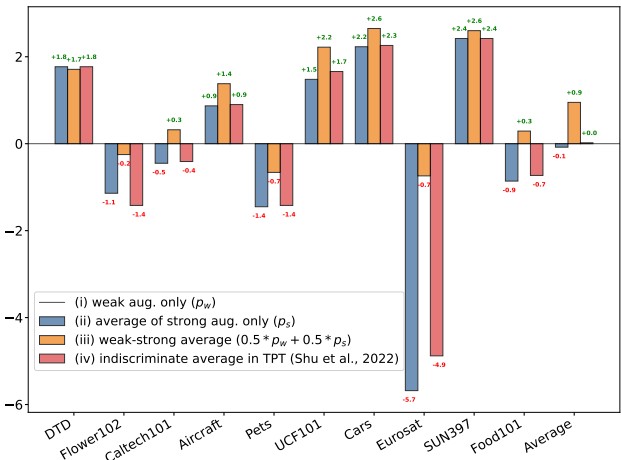

*Figure 2.* Accuracy gains over $p_w(x)$ across fine-grained datasets using the ViT-B/16 backbone under various inference strategies. Ensembling $p_w(x)$ and $p_s(x)$ via a simple average consistently yields more stable and superior performance.

cross-view consistency (Zhang et al., 2022a), TPT updates the prompt learner by minimizing the marginal entropy,

$$\mathcal{L}_{mem} = -\bar{p}_\theta(x) \log \bar{p}_\theta(x). \quad (2)$$

In the inference stage, TPT predicts on $\mathcal{A}_0(x)$ using the updated model $\theta$ to obtain the final prediction as follows,

$$\widehat{p}(x) = p_\theta\left(\mathcal{A}_0\left(x\right)\right). \quad (3)$$

Specifically, we introduce an auxiliary distribution $q$ and reformulate the standard marginal entropy in Eq. (2) as,

$$
\begin{aligned}
\mathcal{L}_{mem} &= -\bar{p}_\theta(x) \log\left(q \cdot \frac{\bar{p}_\theta(x)}{q}\right) \\
&= -\bar{p}_\theta(x) \log q - \bar{p}_\theta(x) \log\left(\frac{\bar{p}_\theta(x)}{q}\right) \quad (4) \\
&= \text{RCE}(\bar{p}_\theta(x); q) + \text{KL}(\bar{p}_\theta(x)\|q).
\end{aligned}
$$

This reformulation reveals that the marginal entropy objective naturally decomposes into a reverse cross-entropy (RCE) term and a Kullback–Leibler (KL) divergence. We define q(x) as stopgrad($\bar{p}_\theta(x)$), which serves as a fixed target distribution. With this choice, the KL term is already optimized and can be dropped, reducing the marginal entropy optimization to the following RCE minimization term,

$$\mathcal{L}_{mem} \approx \text{RCE}(\bar{p}_\theta(x); q(x)). \quad (5)$$

The RCE term can be interpreted as encouraging each augmented view to align its prediction with a fixed target distribution q(x). Prior work has theoretically proven that RCE is noise-tolerant, which has been widely used in noisy label learning (Wang et al., 2019b; Huang et al., 2020). From this perspective (Yoon et al., 2024), q(x) naturally serves as a

pseudo label, closely resembling a self-learning mechanism. Although this process is often described as uncertainty reduction without an explicit training objective (Yoon et al., 2024; Sharifdeen et al., 2025), our analysis reveals that it drives the selected augmented samples to update toward self-generated pseudo labels instead.

### 3.2. Self-Ensembling for Pseudo-Label Estimation

Building on this interpretation above, a key question arises: *how can we obtain a better pseudo label q(x) in Eq.* (5)*?*

In methods based on marginal entropy minimization, represented by TPT (Shu et al., 2022; Feng et al., 2023; Yoon et al., 2024), q(x) is implicitly constructed by averaging predictions across augmented views. However, we find that in some cases, the zero-shot accuracy of weak augmentation is even higher than that of confident augmentations.

Motivated by this, we take a closer look at how weak and strong augmentations differ in practice. We retain the weak augmentation $\mathcal{A}_0(x)$ and select a collection of strong augmentations according to Eq. (1), which together form the set $\mathcal{S}$. For clarity, we denote the weak prediction as $p_w(x) = p_\theta(\mathcal{A}_0(x))$, and the average over selected strong augmentations as $p_s(x) = \frac{1}{|\mathcal{S}|-1} \sum_{z \in \mathcal{S} \setminus \{\mathcal{A}_0(x)\}} p_\theta(z)$.

We then conduct a preliminary study using four inference strategies: (i) using the weak prediction $p_w(x)$ only; (ii) using the strong prediction $p_s(x)$ only, (iii) averaging the weak prediction $p_w(x)$ and the strong prediction $p_s(x)$, and (iv) indiscriminately averaging predictions from selected augmentations, as adopted in TPT. As shown in Fig. 2, the effectiveness of different data augmentations varies substantially across datasets. In some datasets, such as SUN397, strong augmentations yield clear performance gains, while in others, the weak augmentation is more reliable, particularly on EuroSAT. However, existing approaches, including TPT, struggle to consistently provide high-quality pseudo labels in this scenario. In contrast, a simple average of weak and strong augmentations produces more accurate and stable pseudo labels across diverse settings, which is evident on datasets such as Caltech101, Aircraft, and Food101.

Based on this observation, we design a self-ensembling (SE) strategy that dynamically adjusts the ensembling weights around a uniform average, based on the quality of the weak augmentation. Intuitively, if the weak augmentation yields lower entropy than the majority of strong augmentations, it is considered more reliable and should be assigned a larger weight. To quantify it, we compute $\delta$, which represents the relative self-entropy rank of the weakly augmented view $\mathcal{A}_0(x)$ among the strong augmented set $\{\mathcal{A}_i(x)\}_{i=1}^{N-1}$:

$$\delta = \frac{1}{N-1} \sum_{i=1}^{N-1} \mathbb{I}\left[\mathbf{H}(p_\theta(\mathcal{A}_i(x))) > \mathbf{H}(p_\theta(\mathcal{A}_0(x)))\right]. \quad (6)$$

To prevent weight collapse at extreme values (*i.e.*, 0 or 1), which would result in one view being entirely discarded, we rescale $\delta$ into $\beta$ centered at $0.5$ using a linear function. Specifically, $\beta = 0.5 + \gamma * (\delta - 0.5)$, where $\gamma$ controls the dynamic width. We fix $\gamma = 0.4$ across all experiments, constraining $\beta$ within the range $[0.3, 0.7]$. Further discussion of this parameter can be found in Appendix D. Finally, the ensembled prediction $q(x)$ is formulated as:

$$\mathrm{q}(x) = \beta \cdot p_{\mathrm{w}}(x) + (1 - \beta) \cdot p_{\mathrm{s}}(x). \tag{7}$$

### 3.3. Unified Self-Ensembling Framework

As elaborated above, SE emphasizes the test image itself, resulting in more reliable and stable pseudo-labels. Building upon this, we optimize all samples in $\mathcal{S}$ during training, leading to the new training objective as follows,

$$
\begin{aligned}
\mathcal{L}_{rce} &= \mathrm{RCE}(\bar{p}_\theta(\mathbf{x}); \mathrm{q}(x)) \\
&= \mathrm{RCE}(\frac{1}{|\mathcal{S}|} \sum_{z \in \mathcal{S}} p_\theta(z); \mathrm{q}(x)).
\end{aligned} \tag{8}
$$

To reduce computational cost, we introduce a skip technique that disables optimization when all predictions in $\mathcal{S}$ are consistent. After updating the model, we design an inference strategy that explicitly aligns with the training objective. In contrast to the observations in Eq. (3), existing methods (Shu et al., 2022; Yoon et al., 2024) perform prediction solely on the test image $\mathcal{A}_0(x)$, thereby overlooking predictions under strong augmentations. Therefore, we further exploit them at inference time through SE to get the final prediction:

$$\widehat{p}(x) = \beta \cdot p_{\mathrm{w}}(x) + (1 - \beta) \cdot p_{\mathrm{s}}(x). \tag{9}$$

Note that, $p_{\mathrm{w}}(x)$ and $p_{\mathrm{s}}(x)$ are computed using the same augmentation views from $\mathcal{S}$, but with the updated model $\theta$. Together, we have presented all the essential components of USE, which ensures consistency between the training and inference objectives via the SE strategy. Moreover, this ensemble strategy can be directly applied to existing TTA methods (Shu et al., 2022; Sheng et al., 2025b; Dafnis & Metaxas, 2025; Yoon et al., 2024), serving as an optimization-free method on its own. The pseudocode can be found in Alg. 1.

## 4. Experiments

### 4.1. Setup

**Datasets.** We evaluate the proposed USE framework under both ImageNet (Deng et al., 2009) and its out-of-distribution (OOD) variants and fine-grained datasets settings. Specifically, **ImageNet-A** (Hendrycks et al., 2021b) focuses on naturally occurring hard examples, **ImageNet-V2** (Recht et al., 2019) includes large-scale natural images, **ImageNet-R** (Hendrycks et al., 2021a) consists of artistic style variations, and **ImageNet-Sketch** (Wang et al., 2019a) contains

---

**Algorithm 1** Unified Self-Ensembling Framework (USE)

**Require:** Test image $x$, model parameters $\theta$, augmentation function $\mathcal{A}$, step size $\eta$, number of augmentations $N$
**Ensure:** Final prediction $\widehat{p}(x)$
 1: Generate weak augmentation $\mathcal{A}_0(x)$
 2: **for** $i = 1$ to $N - 1$ **do**
 3:     Generate strong augmentation $\mathcal{A}_i(x)$
 4:     Generate prediction $p_\theta(\mathcal{A}_i(x))$
 5:     Compute entropy $\mathbf{H}(p_\theta(\mathcal{A}_i(x)))$
 6: **end for**
 7: # Estimate pseudo-label with SE
 8: Compute $\rho$-th percentile threshold $\tau$
 9: Compute relative self-entropy rank $\delta$ using Eq. (6)
10: Initialize reliable set $\mathcal{S} \leftarrow \{\mathcal{A}_0(x)\}$
11: $\mathcal{S} \leftarrow \mathcal{S} \cup \{\mathcal{A}_i(x) \mid \mathbf{H}(p_\theta(\mathcal{A}_i(x))) < \tau\}$
12: Compute pseudo label $\mathrm{q}(x)$ using Eq. (7)
13: # Optimize text prompt using RCE
14: Compute $\mathcal{L}_{rce}$ using Eq. (8)
15: Update parameters $\theta \leftarrow \theta - \eta \nabla_\theta \mathcal{L}_{rce}$
16: # Reapply SE to generate final prediction
17: Recompute $p_{\mathrm{w}}(x)$ and $p_{\mathrm{s}}(x)$
18: Compute final prediction $\widehat{p}(x)$ using Eq. (9).

---

black and white sketch-style drawings. Beyond natural distribution shifts benchmark, we further evaluate our method on fine-grained datasets, including **DTD** (Cimpoi et al., 2014), **Flowers102** (Nilsback & Zisserman, 2008), **Caltech101** (Fei-Fei et al., 2004), **Aircraft** (Maji et al., 2013), **Pets** (Parkhi et al., 2012), **UCF101** (Soomro et al., 2012), **Cars** (Krause et al., 2013), **EuroSAT** (Helber et al., 2019), **SUN397** (Xiao et al., 2016) and **Food101** (Bossard et al., 2014) , covering a wide range of generalization scenarios.

**Baselines.** We compare our methods with existing optimization-free and optimization-based TTA methods. Under the optimization-free setting, we evaluate SE against CLIP (Radford et al., 2021), MTA (Zanella & Ben Ayed, 2024b) and ZERO (Farina et al., 2024). Under the optimization-based setting, we compare USE with TPT (Shu et al., 2022), C-TPT (Yoon et al., 2024), RLCF (Zhao et al., 2024), TTL (Imam et al., 2025), TPS (Sui et al., 2025), R-TPT (Sheng et al., 2025b), and STS (Dafnis & Metaxas, 2025). All reported results are reproduced using the benchmark provided in (Sheng et al., 2025a).

**Implementation details.** We report results using two standard vision backbones, ViT-B/16 and ResNet-50. For textual prompts, we adopt a context length of 4 with the default template "a photo of a". Each test image is augmented 63 times using AugMix (Hendrycks et al., 2020), from which the top 10% most confident samples ($\rho = 0.1$) are selected. Optimization is performed using the AdamW optimizer with a learning rate of 0.005, and the TTA step is set to 1 as default. All reported results are averaged over runs with different

*Table 1.* Classification accuracy (%) on ImageNet and OOD variants datasets using the ViT-B/16 backbone. Bold indicates the best results.

| Method | Optim. | ImageNet | -A | -V2 | -R | -Sketch | Avg. | OOD Avg. |
|---|---|---|---|---|---|---|---|---|
| CLIP | × | 66.72 | 47.83 | 60.94 | 73.99 | 46.10 | 59.12 | 57.22 |
| MTA (Zanella & Ben Ayed, 2024b) | × | 69.26 | 57.10 | 63.63 | 76.96 | 48.45 | 63.08 | 61.54 |
| ZERO (Farina et al., 2024) | × | **69.31** | **59.79** | **64.20** | 77.28 | 48.41 | **63.80** | **62.42** |
| SE | × | 69.15 | 59.15 | 64.10 | **77.34** | **48.53** | 63.65 | 62.28 |
| TPT (Shu et al., 2022) | ✓ | 68.92 | 54.59 | 63.45 | 77.06 | 47.84 | 62.37 | 60.74 |
| C-TPT (Yoon et al., 2024) | ✓ | 68.45 | 51.15 | 62.66 | 75.78 | 47.42 | 61.09 | 59.25 |
| RLCF (Zhao et al., 2024) | ✓ | 68.55 | 57.48 | 63.42 | 77.10 | 48.07 | 62.92 | 61.52 |
| TTL (Imam et al., 2025) | ✓ | 69.32 | 58.84 | 64.28 | 77.66 | 48.73 | 63.77 | 62.38 |
| TPS (Sui et al., 2025) | ✓ | 68.82 | 57.93 | 63.45 | 76.95 | 48.06 | 63.04 | 61.60 |
| R-TPT (Sheng et al., 2025b) | ✓ | 69.31 | 57.61 | 63.98 | 76.90 | 47.70 | 63.10 | 61.55 |
| STS (Dafnis & Metaxas, 2025) | ✓ | 68.77 | **61.37** | 64.20 | 77.02 | 48.09 | 63.89 | 62.67 |
| USE | ✓ | **69.72** | 59.79 | **64.60** | **77.89** | **48.91** | **64.18** | **62.80** |

*Table 2.* Classification accuracy (%) on fine-grained datasets using the ViT-B/16 backbone.

| Method | Optim. | DTD | Flower102 | Caltech101 | Aircraft | Pets | UCF101 | Cars | EuroSAT | SUN397 | Food101 | Avg. |
|---|---|---|---|---|---|---|---|---|---|---|---|---|
| CLIP | × | 44.33 | 67.32 | 93.95 | 23.85 | **88.20** | 65.19 | 65.56 | 42.05 | 62.58 | 83.66 | 63.67 |
| MTA (Zanella & Ben Ayed, 2024b) | × | 46.07 | **67.34** | 94.20 | 24.65 | 88.05 | **67.76** | 67.61 | 42.33 | 65.26 | 84.40 | 64.77 |
| ZERO (Farina et al., 2024) | × | 45.36 | 66.91 | 93.87 | 24.72 | 87.33 | 66.60 | 67.27 | 37.19 | **65.51** | 83.80 | 63.86 |
| SE | × | **46.45** | 66.91 | **94.36** | **25.13** | 87.86 | 67.75 | **67.86** | 44.44 | 65.02 | **84.44** | **65.02** |
| TPT (Shu et al., 2022) | ✓ | 47.05 | 68.64 | 94.06 | 23.30 | 87.30 | 68.38 | 66.59 | 42.90 | 65.51 | 84.64 | 64.84 |
| C-TPT (Yoon et al., 2024) | ✓ | 45.09 | **69.65** | 93.69 | 24.05 | **88.20** | 65.03 | 65.81 | 42.44 | 64.46 | 83.14 | 64.16 |
| RLCF (Zhao et al., 2024) | ✓ | 46.69 | 67.54 | **94.56** | 22.25 | 86.74 | 67.46 | 66.66 | **43.40** | 65.11 | 84.18 | 64.46 |
| TTL (Imam et al., 2025) | ✓ | 45.86 | 66.26 | 93.75 | 24.45 | 87.16 | 67.06 | 66.47 | 39.26 | 65.17 | 83.90 | 63.93 |
| TPS (Sui et al., 2025) | ✓ | 45.66 | 67.32 | 94.00 | 24.71 | 87.50 | 67.37 | 67.69 | 42.60 | 64.64 | 84.45 | 64.59 |
| R-TPT (Sheng et al., 2025b) | ✓ | 46.36 | 68.25 | 93.81 | 23.75 | 86.94 | 67.59 | 66.85 | 35.07 | 65.45 | 84.26 | 63.83 |
| STS (Dafnis & Metaxas, 2025) | ✓ | 46.19 | 65.89 | 93.57 | **24.77** | 86.64 | 66.89 | 67.14 | 38.35 | 64.88 | 83.04 | 63.74 |
| USE | ✓ | **47.61** | 68.78 | 94.50 | 24.35 | 87.94 | **68.68** | 67.72 | 43.31 | **65.89** | **84.71** | **65.35** |

random seeds. For brevity, *we denote ImageNet and its OOD variants datasets as IN&O, and fine-grained datasets as FGVC*, and provide their average. Detailed results are provided in Appendix F and G.

## 4.2. Main Results

**Results on ImageNet and OOD variants.** We evaluate the generalization performance of SE and USE on ImageNet and its OOD variants. We compare top-1 accuracy against prior methods using ViT-B/16 and ResNet-50 backbones, with results reported in Table 1 and Table 3, respectively. For brevity, we abbreviate optimization as Optim. in all tables, and × indicates that a method is optimization-free. Despite its simplicity, SE achieves the best or second-best results across various settings among optimization-free approaches. Specifically, it obtains an average accuracy of 47.80% on the ResNet50 backbone, even outperforming most optimization-based methods. Although STS (Dafnis & Metaxas, 2025) slightly outperforms USE on the ImageNet-A dataset, which contains substantial compromised examples that our method prioritizes. USE still yields superior performance across the majority of datasets and maintains a clear margin in terms of the average accuracy.

**Results on fine-grained datasets.** We report image clas-

sification accuracy on ten fine-grained datasets using two vision backbones, with results summarized in Table 2 and Table 4. Overall, both USE and SE achieve the best average performance across the datasets. Notably, SE attains an accuracy of 44.44% on EuroSAT with ViT-B/16, yielding a substantial gain of 2.39%, where counterparts like ZERO (Farina et al., 2024) even degrades compared to zero-shot performance. By applying a consistent framework, USE successfully enhances its TPT (Shu et al., 2022) baseline, raising the average accuracy from 64.84% to 65.35% with the ViT-B/16 backbone.

**Initialization with CoOp.** Following prior work (Shu et al., 2022), we further investigate the robustness of our method to different pretrained models by adopting the stronger CoOp initialization (Zhou et al., 2022b) on the ViT-B/16 backbone. The results are reported in Table 5. Both SE and USE achieve the best or second-best performance across all three evaluation settings, demonstrating the robustness of our framework under stronger initialization. In particular, across the fine-grained evaluation, SE attains an average accuracy of 64.57%, significantly outperforming its counterparts. Meanwhile, the optimization-based version USE achieves performance comparable to the best-performing method RLCF (Zhao et al., 2024), while requiring substantially lower computational cost.

*Table 3.* Classification accuracy (%) on ImageNet and OOD variants datasets using the ResNet50 backbone.

| Method | Optim. | ImageNet | -A | -V2 | -R | -Sketch | Avg. | OOD Avg. |
|---|---|---|---|---|---|---|---|---|
| CLIP (Radford et al., 2021) | × | 58.17 | 21.87 | 51.45 | 56.12 | 33.35 | 44.19 | 40.70 |
| MTA (Zanella & Ben Ayed, 2024b) | × | **60.43** | 27.63 | 54.43 | 58.49 | 35.31 | 47.26 | 43.97 |
| ZERO (Farina et al., 2024) | × | 60.29 | 30.00 | **54.59** | 57.80 | 34.77 | 47.49 | 44.29 |
| SE | × | 60.39 | **30.19** | 54.30 | **58.64** | **35.48** | **47.80** | **44.65** |
| TPT (Shu et al., 2022) | ✓ | 60.63 | 26.51 | 54.64 | 59.03 | 35.14 | 47.19 | 43.83 |
| C-TPT (Yoon et al., 2024) | ✓ | 60.47 | 24.27 | 54.21 | 57.73 | 34.72 | 46.28 | 42.73 |
| RLCF (Zhao et al., 2024) | ✓ | 60.14 | 28.56 | 54.22 | 58.56 | 35.13 | 47.32 | 44.12 |
| TPS (Sui et al., 2025) | ✓ | 59.93 | 28.32 | 53.91 | 58.32 | 34.95 | 47.09 | 43.88 |
| R-TPT (Sheng et al., 2025b) | ✓ | 60.76 | 28.11 | 54.64 | 57.70 | 34.05 | 47.05 | 43.63 |
| STS (Dafnis & Metaxas, 2025) | ✓ | 59.69 | **32.15** | 53.88 | 57.51 | 34.69 | 47.58 | 44.56 |
| USE | ✓ | **61.12** | 30.71 | **55.16** | **59.23** | **35.84** | **48.41** | **45.24** |

*Table 4.* Classification accuracy (%) on fine-grained datasets using the ResNet50 backbone.

| Method | Optim. | DTD | Flower102 | Caltech101 | Aircraft | Pets | UCF101 | Cars | EuroSAT | SUN397 | Food101 | Avg. |
|---|---|---|---|---|---|---|---|---|---|---|---|---|
| CLIP (Radford et al., 2021) | × | 40.37 | **61.67** | 85.88 | 15.72 | 83.54 | 58.87 | 55.78 | **23.65** | 58.84 | 73.94 | 55.83 |
| MTA (Zanella & Ben Ayed, 2024b) | × | 39.92 | 60.98 | 87.46 | **17.94** | 84.79 | **60.52** | **58.74** | 22.39 | 60.75 | 74.32 | 56.78 |
| ZERO (Farina et al., 2024) | × | 39.86 | 59.07 | 86.43 | 17.78 | 84.26 | 59.13 | 58.00 | 21.88 | 60.46 | 72.27 | 55.91 |
| SE | × | **40.54** | 61.29 | **87.75** | 17.31 | **84.93** | 60.49 | 58.73 | **23.65** | **60.76** | **74.47** | **56.99** |
| TPT (Shu et al., 2022) | ✓ | 41.55 | 62.61 | 87.77 | 17.67 | 84.46 | 60.59 | 58.57 | **28.43** | 61.38 | **75.01** | 57.80 |
| C-TPT (Yoon et al., 2024) | ✓ | 41.34 | **65.04** | 87.34 | 17.18 | 83.70 | 60.27 | 56.40 | 27.22 | 60.94 | 74.83 | 57.43 |
| RLCF (Zhao et al., 2024) | ✓ | 41.61 | 60.27 | 87.83 | 16.92 | 83.35 | 60.76 | 57.88 | 26.73 | 60.67 | 74.08 | 57.01 |
| TPS (Sui et al., 2025) | ✓ | 40.25 | 61.12 | 86.96 | 17.37 | 84.66 | 60.60 | 58.42 | 24.48 | 60.36 | 74.38 | 56.86 |
| R-TPT (Sheng et al., 2025b) | ✓ | 41.31 | 61.31 | 86.13 | 17.58 | 84.19 | 59.37 | 58.39 | 21.27 | 60.79 | 73.41 | 56.38 |
| STS (Dafnis & Metaxas, 2025) | ✓ | 39.63 | 57.98 | 86.63 | 17.31 | 83.48 | 59.30 | 57.58 | 22.12 | 59.91 | 71.35 | 55.53 |
| USE | ✓ | **41.84** | 62.89 | **87.95** | **17.85** | **85.32** | **61.30** | **58.99** | 25.65 | **61.55** | 74.86 | **57.82** |

*Table 5.* Classification accuracy (%) with CoOp initialization on the ViT-B/16 backbone.

| Method | Optim. | IN&O | FGVC | Avg. |
|---|---|---|---|---|
| CoOp (Zhou et al., 2022b) | × | 62.20 | 63.27 | 62.73 |
| MTA (Zanella & Ben Ayed, 2024b) | × | 65.76 | 64.14 | 64.95 |
| ZERO (Farina et al., 2024) | × | **66.38** | 63.81 | 65.10 |
| SE | × | 66.33 | **64.57** | **65.45** |
| TPT (Shu et al., 2022) | ✓ | 65.21 | 64.78 | 64.99 |
| C-TPT (Yoon et al., 2024) | ✓ | 63.86 | 64.30 | 64.08 |
| RLCF (Zhao et al., 2024) | ✓ | 65.40 | **65.00** | 65.20 |
| TTL (Imam et al., 2025) | ✓ | 66.09 | 63.99 | 65.04 |
| TPS (Sui et al., 2025) | ✓ | 66.00 | 64.34 | 65.17 |
| R-TPT (Sheng et al., 2025b) | ✓ | 65.55 | 64.06 | 64.80 |
| STS (Dafnis & Metaxas, 2025) | ✓ | **66.45** | 63.30 | 64.88 |
| USE | ✓ | 66.15 | 64.92 | **65.53** |

*Table 6.* Ablation study on the ViT-B/16 backbone, evaluating the impact of different optimization and inference strategies.

| Optim. | Inference | IN&O | FGVC | Avg. |
|---|---|---|---|---|
| - | standard | 59.12 | 63.67 | 61.39 |
| - | uniform | **64.45** | 63.95 | 64.20 |
| - | SE | 63.65 | **65.02** | **64.34** |
| MEM | standard | 62.37 | **64.84** | 63.60 |
| MEM | uniform | **64.26** | 64.03 | 64.14 |
| MEM | SE | **64.26** | 64.60 | **64.43** |
| RCE | standard | 62.13 | 65.30 | 63.71 |
| RCE | uniform | **64.45** | 64.68 | 64.56 |
| RCE | SE | 64.18 | **65.35** | **64.77** |

### 4.3. Ablation Study

We conduct ablation studies on the ViT-B/16 backbone to analyze the contribution of individual components in our approach, including the optimization objective and the inference strategy. The results are summarized in Table 6. Standard denotes using weak augmentation only, while uniform denotes averaging the predictions of selected augmentations equally. Each component contributes positively to zero-shot performance, but the primary gain stems from applying SE at inference. During optimization, while the RCE objective

alone performs comparably to MEM, combining it with SE yields superior results, highlighting their complementarity. During inference, substituting SE with alternative strategies degrades performance, such as uniform, demonstrating the importance of maintaining consistent objectives. Finally, the USE framework delivers the best overall performance, achieving an accuracy of 64.77%.

### 4.4. Further Analysis

**Analysis of SE strategy.** To further examine the generalization ability of SE, we apply it to representative TTA methods, including TPT (Shu et al., 2022), C-TPT (Yoon

*Table 7.* Classification accuracy (%) on ViT-B/16 when integrating SE as a plug-in module into representative TTA methods.

| Method | IN&O | FGVC | Avg. |
|---|---|---|---|
| TPT (Shu et al., 2022) | 62.37 | **64.84** | 63.60 |
| + **SE** | **64.26** | 64.60 | **64.43** |
| C-TPT (Yoon et al., 2024) | 61.09 | 64.16 | 62.62 |
| + **SE** | **64.27** | **64.60** | **64.43** |
| R-TPT (Sheng et al., 2025b) | 63.10 | 63.83 | 63.47 |
| + **SE** | **64.26** | **63.85** | **64.06** |
| STS (Dafnis & Metaxas, 2025) | **63.89** | 63.74 | 63.81 |
| + **SE** | 63.82 | **64.97** | **64.40** |

*Table 8.* Classification accuracy (%) on the ViT-B/16 backbone for different choices of the rescaling function in SE.

| Function | IN&O | FGVC | Avg. |
|---|---|---|---|
| $\beta = 0.5 \pm \gamma \cdot \sqrt{|\delta - 0.5|}$ | 63.38 | 65.01 | 64.19 |
| $\beta = 0.5 \pm \gamma \cdot (\delta - 0.5)^2$ | **63.81** | 64.94 | **64.38** |
| $\beta = 0.5 + \gamma \cdot (\delta - 0.5)$ (**default**) | 63.65 | **65.02** | 64.34 |

*Table 9.* Classification accuracy (%) on the ViT-B/16 backbone for representative TTA methods using text prompt ensembles.

| Method | IN&O | FGVC | Avg. |
|---|---|---|---|
| CLIP (Radford et al., 2021) | 61.23 | 64.50 | 62.87 |
| ZERO (Farina et al., 2024) | 65.75 | 64.70 | 65.22 |
| TPS (Sui et al., 2025) | 65.22 | 65.31 | 65.26 |
| STS (Dafnis & Metaxas, 2025) | **65.91** | 64.40 | 65.16 |
| **SE** | 65.73 | **65.49** | **65.61** |

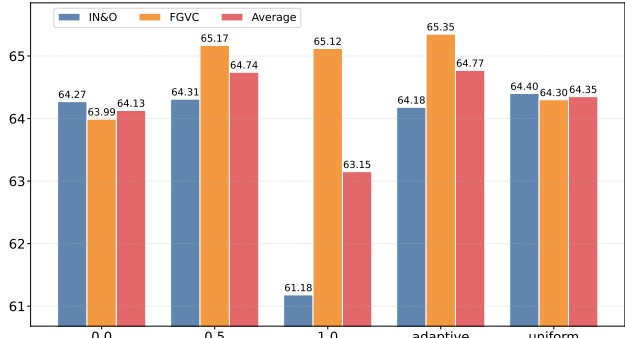

*Figure 3.* Classification accuracy (%) under different ensembling strategies using the ViT-B/16 backbone. Specifically, "Adaptive" refers to SE strategy, which adjusts weights dynamically.

et al., 2024), R-TPT (Sheng et al., 2025b), and STS (Dafnis & Metaxas, 2025). As reported in Table 7, incorporating SE consistently enhances performance, yielding clear gains in average accuracy. Notably, SE increases the accuracy of C-TPT (Yoon et al., 2024) from 61.09% to 64.27% in IN&O datasets, demonstrating its robustness as a simple yet effective plug-in strategy.

**Analysis of hyperparameter sensitivity.** We evaluate the sensitivity of USE to the ensembling factor $\beta$, as illustrated in Figure 3. Specifically, we compare the adaptive SE strategy against fixed choices of $\beta \in \{0.0, 0.5, 1.0\}$, as well as a uniform alternative ($\beta = \frac{1}{|\mathcal{S}|}$). The results demonstrate that applying SE within the USE framework maintains stable performance on the IN&O benchmark while achieving the highest accuracy on FGVC datasets. Ultimately, SE delivers the best overall performance, validating the effectiveness of the adaptive self-ensembling strategy.

**Analysis of rescaling function.** We also investigate more complex variants beyond the linear mapping to adaptively adjust $\beta$ in Eq. (7). Specifically, we explore a square-root formulation ($\beta = 0.5 \pm \gamma \cdot \sqrt{|\delta - 0.5|}$) and a squared formulation ($\beta = 0.5 \pm \gamma \cdot (\delta - 0.5)^2$), and evaluate them within SE. As shown in Table 8, although these non-linear alternatives can yield performance gains, we adopt the linear

design for its balance of simplicity and effectiveness.

**Analysis of text prompt ensembles.** Beyond the generally applied prompt template, we further investigate SE using an ensemble of multiple text templates. Specifically, following STS (Dafnis & Metaxas, 2025), we employ seven generic prompt templates, which are detailed in Appendix E. Because several TTA methods do not support text ensembling by design, we compare SE exclusively with ZERO (Farina et al., 2024), TPS (Sui et al., 2025), and STS (Dafnis & Metaxas, 2025), with the results reported in Table 9. When integrated with text ensembles, SE enhances CLIP performance from 62.87% to 65.61%, delivering better results even when multiple templates are applied. Although our method (SE) slightly underperforms STS (Dafnis & Metaxas, 2025) on the IN&O datasets, it requires no additional training and still improves upon vanilla CLIP by a notable margin of 4.5%.

**Analysis of computational efficiency.** As shown in Table 10, we compare the computational overhead of SE and USE with their counterparts. The results report testing time per sample, peak memory usage, classification accuracy, and accuracy gains over zero-shot CLIP. SE achieves the lowest time and memory costs among optimization-free TTA methods, yielding a 1.35% accuracy gain with an overhead comparable to 64 forward passes of CLIP. The computational overhead of USE is further reduced by introducing a skip technique, demonstrating an average testing time of 0.144 seconds per image.

Specifically, if the prediction obtained from the weakly augmented view is identical to the selected strong augmentations, the prediction is considered reliable, and subsequent optimization is skipped. We report both the skipping ratio and the resulting accuracy in Figure 4. In practice, over 60% of the test samples are skipped without accuracy degradation observed, reducing computational overhead while

*Table 10.* Efficiency comparison of various TTA methods. "CLIP-64" refers to the computational overhead of 64 forward passes.

| Method | Time (s) | Mem (GB) | Acc (%) | Gains (%) |
|---|---|---|---|---|
| CLIP | 0.017 | 0.853 | 63.67 | - |
| ZERO (Farina et al., 2024) | 0.187 | 1.185 | 63.86 | 0.19 |
| MTA (Zanella & Ben Ayed, 2024b) | 0.069 | 1.060 | 64.77 | 1.10 |
| STS (Dafnis & Metaxas, 2025) | 0.073 | 1.144 | 63.74 | 0.07 |
| **SE** | **0.059** | **1.060** | **65.02** | **1.35** |
| CLIP-64 | 0.059 | 1.060 | 63.67 | - |
| TPT (Shu et al., 2022) | 0.200 | 6.579 | 64.84 | 1.17 |
| R-TPT (Sheng et al., 2025a) | 0.298 | 6.579 | 63.83 | 0.16 |
| C-TPT (Yoon et al., 2024) | 0.202 | 6.579 | 64.16 | 0.49 |
| RLCF (Zhao et al., 2024) | 2.697 | 7.329 | 64.46 | 0.79 |
| TTL (Imam et al., 2025) | 0.180 | 6.473 | 63.93 | 0.26 |
| **USE** | **0.144** | **6.568** | **65.35** | **1.68** |

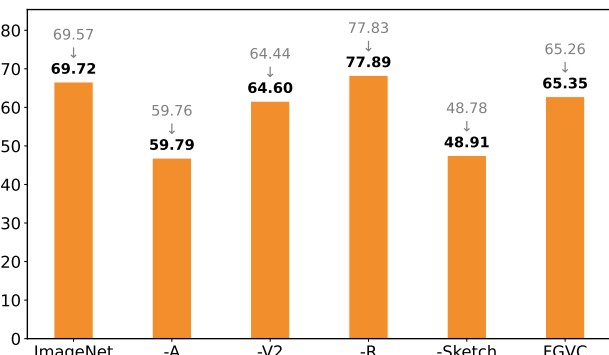

*Figure 4.* Skipping ratio and classification accuracy (%) before/after using the skip technique on the ViT-B/16 backbone. Numbers in gray / **black** denote accuracies before / after skipping.

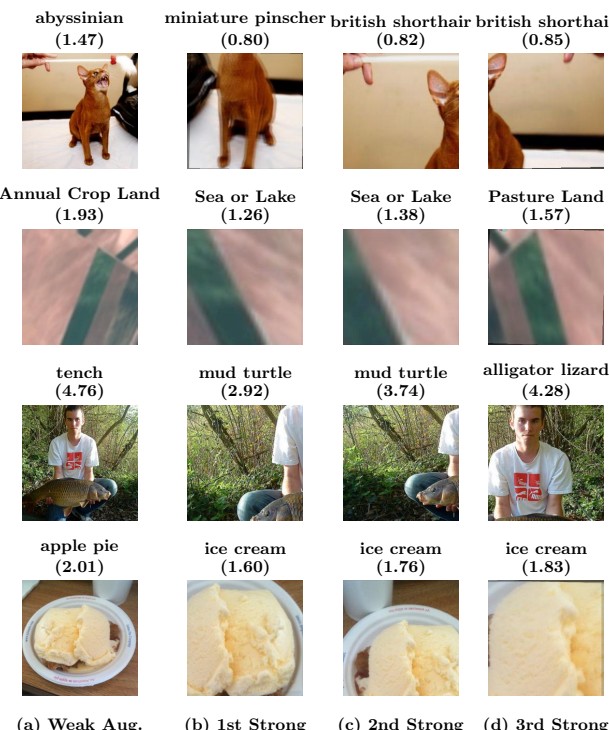

| abyssinian (1.47) | miniature pinscher (0.80) | british shorthair (0.82) | british shorthair (0.85) |
| Annual Crop Land (1.93) | Sea or Lake (1.26) | Sea or Lake (1.38) | Pasture Land (1.57) |
| tench (4.76) | mud turtle (2.92) | mud turtle (3.74) | alligator lizard (4.28) |
| apple pie (2.01) | ice cream (1.60) | ice cream (1.76) | ice cream (1.83) |
| (a) Weak Aug. | (b) 1st Strong | (c) 2nd Strong | (d) 3rd Strong |

*Figure 5.* Visualization of the differences between weak and strong augmentations. Column (a) presents the weak augmentation, while Columns (b)–(d) display the top-3 strong augmentations sorted by entropy. Predictions are obtained using a ViT-B/16 backbone.

preserving performance effectively.

### 4.5. Visualization

To provide a more intuitive understanding of the differences between weak and strong augmentations, we visualize representative examples in Figure 5. Specifically, we present four sets of images sampled from diverse datasets, including ImageNet, Pets, EuroSAT, and Food101. For each image, the predicted label is shown above the image, with the corresponding entropy value reported in parentheses. Notably, we can observe that while weak augmentations sometimes yield relatively higher entropy than strong augmentations, they may still maintain more stable semantic information. By explicitly retaining and prioritizing the original test image, SE provides performance gains consistently.

## 5. Conclusion

This paper revisited the classical TPT approach from a self-learning perspective and introduced a new test-time prompt tuning framework named Unified Self-Ensembling (USE). USE introduces a simple yet effective ensembling strategy that adaptively prioritizes the test image to generate more reliable pseudo-labels during optimization. USE further employs the same Self-Ensembling (SE) strategy to integrate augmented views with the test image, providing a unified process across both stages. Importantly, SE functions independently as an efficient optimization-free TTA method. Empirical results across various architectures and datasets validate the robustness of both USE and SE. Moreover, SE is computationally efficient and can be easily plugged into existing methods to provide consistent performance gains. We hope our framework provides valuable insights for future research in test-time adaptation.

**Limitations.** Despite its effectiveness, our method faces practical constraints. First, similar to most episodic TTA methods, our approach requires generating and processing multiple augmented views (*e.g.*, 64 forward passes per sample). This naturally introduces additional computational overhead compared to zero-shot inference. Second, since our self-ensembling strategy explicitly prioritizes the weakly augmented view, performance can be degraded in failure cases where the original image is severely corrupted.

## Acknowledgment

This work was funded by the National Natural Science Foundation of China under Grants 62276256 and U2441251. We gratefully acknowledge Dr. Lijun Sheng and Dr. Zhengbo Wang for their critical discussions, and the anonymous reviewers for their constructive comments and helpful suggestions that improved this paper.

## Impact Statement

This paper presents work whose goal is to advance the field of Machine Learning. There are many potential societal consequences of our work, none which we feel must be specifically highlighted here.

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

## A. Dataset Details

We evaluate our method on 15 datasets spanning ImageNet and its OOD variants, as well as multiple fine-grained recognition benchmarks. For each dataset, we report the dataset name, number of classes, number of test samples, and a brief task description in Table 11.

*Table 11.* Summary of evaluation datasets. We report the detailed information for each dataset.

| Dataset | Classes | Testing | Task |
|---|---|---|---|
| ImageNet (Deng et al., 2009) | 1,000 | 50,000 | Large-scale recognition across 1,000 general categories. |
| ImageNet-A (Hendrycks et al., 2021b) | 200 | 7,500 | Naturally occurring adversarial images that foil standard models. |
| ImageNet-V2 (Recht et al., 2019) | 1,000 | 10,000 | A new test set matched to the original ImageNet distribution. |
| ImageNet-R (Hendrycks et al., 2021a) | 200 | 30,000 | Recognition under domain shift (art, cartoons, 3D models). |
| ImageNet-Sketch (Wang et al., 2019a) | 1,000 | 50,889 | Shape-based recognition from sketch-style drawings. |
| DTD (Cimpoi et al., 2014) | 47 | 1,692 | Texture classification based on repetitive visual attributes. |
| Flowers102 (Nilsback & Zisserman, 2008) | 102 | 2,463 | Fine-grained classification of 102 flower species. |
| Caltech101 (Fei-Fei et al., 2004) | 100 | 2,465 | 101 categories with varied pose, scale, and background. |
| Aircraft (Maji et al., 2013) | 100 | 3,333 | Fine-grained recognition of visually similar aircraft variants. |
| Pets (Parkhi et al., 2012) | 37 | 3,669 | Fine-grained classification of 37 cat and dog breeds. |
| UCF101 (Soomro et al., 2012) | 101 | 3,783 | Action recognition across 101 human activity categories. |
| Cars (Krause et al., 2013) | 196 | 8,041 | Fine-grained recognition of 196 distinct car models. |
| EuroSAT (Helber et al., 2019) | 10 | 8,100 | Land-use classification from multi-spectral satellite imagery. |
| SUN397 (Xiao et al., 2016) | 397 | 19,850 | Large-scale classification of 397 indoor/outdoor scenes. |
| Food101 (Bossard et al., 2014) | 101 | 30,300 | Large-scale recognition of 101 popular food categories. |

## B. Empirical Approximation

In this section, we further empirically verify that the Reverse Cross-Entropy objective in Eq. (5) approximates marginal entropy minimization in terms of the accuracy.

As shown in Tables 12 and 13, the classification accuracy on ImageNet and its OOD variants, as well as on fine-grained benchmarks, is nearly identical under the two objectives. The differences are consistently negligible across all datasets, indicating that optimizing marginal entropy or Reverse Cross-Entropy leads to almost the same performance in practice. These results empirically support the claim before, despite their different gradient formulations.

*Table 12.* Classification accuracy using different optimization objectives across ImageNet and OOD variants on ViT-B/16.

| Loss | ImageNet | -A | -V2 | -R | -Sketch | Avg. |
|---|---|---|---|---|---|---|
| MEM | 68.92 | 54.59 | 63.45 | 77.06 | 47.84 | 62.37 |
| RCE | 68.91 | 54.59 | 63.46 | 77.05 | 47.84 | 62.37 |

*Table 13.* Classification accuracy using different optimization objectives across fine-grained datasets on ViT-B/16.

| Loss | DTD | Flower102 | Caltech101 | Aircraft | Pets | UCF101 | Cars | EuroSAT | SUN397 | Food101 | Avg. |
|---|---|---|---|---|---|---|---|---|---|---|---|
| MEM | 47.05 | 68.64 | 94.06 | 23.30 | 87.30 | 68.38 | 66.59 | 42.90 | 65.51 | 84.64 | 64.84 |
| RCE | 47.02 | 68.64 | 94.00 | 23.31 | 87.31 | 68.39 | 66.57 | 42.94 | 65.52 | 84.63 | 64.83 |

## C. Results under Different TTA Steps

We determine the number of optimization steps from {1, 2, 4, 6} across three different datasets, with the results summarized in Fig. 6. The orange line represents USE, while the blue line denotes TPT (Shu et al., 2022). Consistent with the observations

reported in TPT, increasing the number of optimization steps does not always lead to a significant performance improvement. However, USE consistently outperforms TPT across various step settings, demonstrating its effectiveness and robustness.

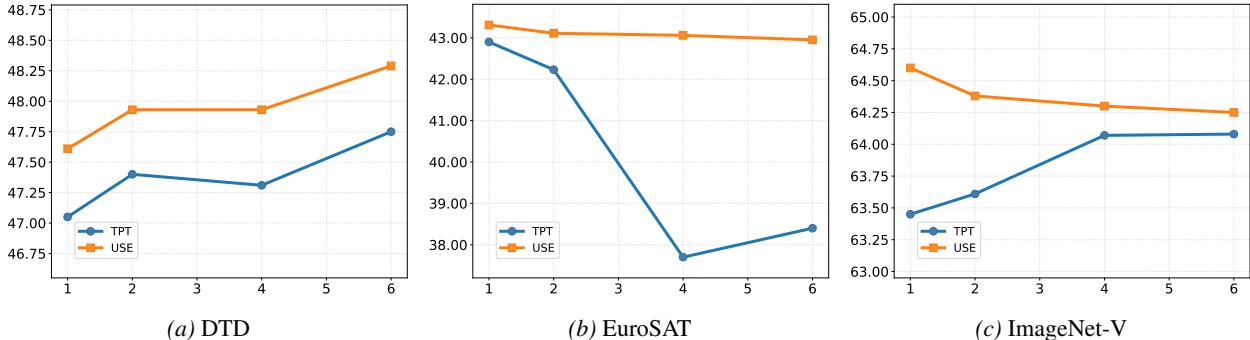

*Figure 6.* Classification accuracy (%) with different TTA steps on ViT-B/16.

## D. Different Scaling Factors $\gamma$

While the optimal scaling factor $\gamma$ varies across domains, empirical results show that fixing $\gamma = 0.4$ yields highly stable performance. As Tables 1–4 demonstrate, this default configuration enables both SE and USE to achieve highly competitive performance across multiple architectures and benchmarks without requiring any domain-specific tuning. Furthermore, we evaluate different choices of $\gamma \in \{0.2, 0.4, 0.6, 0.8, 1.0\}$ on both IN&O and FGVC benchmarks, as shown in Fig. 7a and Fig. 7b.

For the ViT-B/16 backbone, varying $\gamma$ yields stable performance on FGVC datasets, while fixing $\gamma = 0.4$ prevents the performance degradation observed on IN&O when larger scaling factors are used. Specifically, we note that even under our worst-case configuration ($\gamma = 1$), USE still achieves an average accuracy of 64.64% across IN&V and FGVC, outperforming the standard baseline of 63.15% ($\beta = 1$) by a clear margin. Overall, our method demonstrates strong robustness to the choice of $\gamma$, validating the effectiveness of the proposed rescaling strategy.

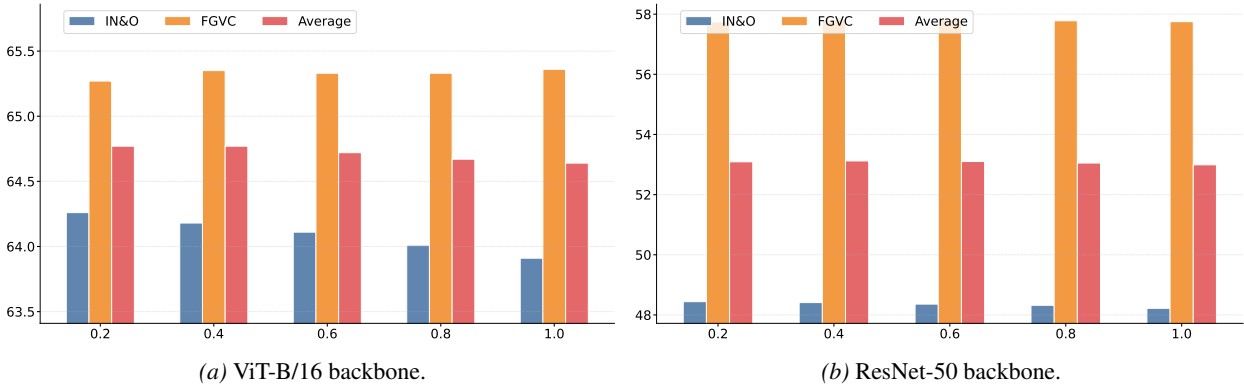

*Figure 7.* Classification accuracy (%) under different rescaling factor $\gamma \in \{0.2, 0.4, 0.6, 0.8, 1.0\}$ on different backbones.

## E. Text Ensembles

Beyond the commonly used prompt "a photo of a {}", we adopt seven generic text templates from the CLIP repository (Radford et al., 2021) to construct text ensembles, as described in Section 4.4. The details of the prompt templates we use are provided in Table 14.

*Table 14.* Seven text templates we used in text ensembles

| Text Templates |
| --- |
| "a bad photo of the { }." |
| "a { } in a video game." |
| "a origami { }." |
| "a photo of the small { }." |
| "art of the { }." |
| "a photo of the large { }." |
| "itap of a { }." |

## F. Standard Deviations

Since Section 4 reports only accuracy results, we additionally provide an analysis based on the standard deviation across multiple runs. The results are summarized in Tables 15, 17, 16, and 18.

As shown in Table 15, SE exhibits slightly larger deviations on ImageNet and OOD benchmarks compared to other optimization-free methods. This behavior is expected, as SE relies exclusively on test-time augmentation, which is the primary source of randomness across different runs. Despite this increased variance, SE achieves the second-best performance in this setting, indicating that it remains simple yet effective. In addition, USE consistently demonstrates smaller standard deviations, reflecting its stable performance across runs.

*Table 15.* Standard deviations on ImageNet and OOD variants datasets using the ViT-B/16 backbone.

| Method | Optim. | ImageNet | -A | -V2 | -R | -Sketch |
| --- | --- | --- | --- | --- | --- | --- |
| MTA (Zanella & Ben Ayed, 2024b) | × | ±0.00 | ±0.09 | ±0.04 | ±0.03 | ±0.03 |
| ZERO (Farina et al., 2024) | × | ±0.04 | ±0.01 | ±0.02 | ±0.06 | ±0.06 |
| SE | × | ±0.06 | ±0.18 | ±0.16 | ±0.04 | ±0.04 |
| TPT (Shu et al., 2022) | ✓ | ±0.05 | ±0.06 | ±0.17 | ±0.05 | ±0.00 |
| C-TPT (Yoon et al., 2024) | ✓ | ±0.01 | ±0.13 | ±0.16 | ±0.07 | ±0.02 |
| RLCF (Zhao et al., 2024) | ✓ | ±0.05 | ±0.17 | ±0.14 | ±0.08 | ±0.06 |
| TTL (Imam et al., 2025) | ✓ | ±0.04 | ±0.16 | ±0.12 | ±0.11 | ±0.02 |
| TPS (Sui et al., 2025) | ✓ | ±0.02 | ±0.19 | ±0.03 | ±0.01 | ±0.06 |
| R-TPT (Sheng et al., 2025b) | ✓ | ±0.05 | ±0.20 | ±0.04 | ±0.03 | ±0.07 |
| STS (Dafnis & Metaxas, 2025) | ✓ | ±0.06 | ±0.31 | ±0.05 | ±0.06 | ±0.02 |
| USE | ✓ | ±0.04 | ±0.19 | ±0.08 | ±0.05 | ±0.03 |

*Table 16.* Standard deviations on ImageNet and OOD variants datasets using the RN-50 backbone.

| Method | Optim. | ImageNet | -A | -V2 | -R | -Sketch |
| --- | --- | --- | --- | --- | --- | --- |
| MTA (Zanella & Ben Ayed, 2024b) | × | ±0.01 | ±0.15 | ±0.14 | ±0.07 | ±0.09 |
| ZERO (Farina et al., 2024) | × | ±0.07 | ±0.05 | ±0.08 | ±0.05 | ±0.01 |
| SE | × | ±0.00 | ±0.18 | ±0.23 | ±0.01 | ±0.01 |
| TPT (Shu et al., 2022) | ✓ | ±0.01 | ±0.23 | ±0.00 | ±0.03 | ±0.03 |
| C-TPT (Yoon et al., 2024) | ✓ | ±0.10 | ±0.11 | ±0.05 | ±0.02 | ±0.01 |
| RLCF (Zhao et al., 2024) | ✓ | ±0.01 | ±0.09 | ±0.11 | ±0.12 | ±0.03 |
| TPS (Sui et al., 2025) | ✓ | ±0.00 | ±0.21 | ±0.16 | ±0.05 | ±0.04 |
| R-TPT (Sheng et al., 2025b) | ✓ | ±0.03 | ±0.19 | ±0.04 | ±0.01 | ±0.00 |
| STS (Dafnis & Metaxas, 2025) | ✓ | ±0.02 | ±0.07 | ± 0.28 | ±0.10 | ±0.02 |
| USE | ✓ | ±0.04 | ±0.25 | ±0.12 | ±0.02 | ±0.00 |

*Table 17.* Standard deviations on fine-grained datasets using the ViT-B/16 backbone.

| Method | Optim. | DTD | Flower102 | Caltech101 | Aircraft | Pets | UCF101 | Cars | EuroSAT | SUN397 | Food101 |
|---|---|---|---|---|---|---|---|---|---|---|---|
| MTA (Zanella & Ben Ayed, 2024b) | × | ±0.15 | ±0.30 | ±0.16 | ±0.22 | ±0.12 | ±0.01 | ±0.02 | ±0.07 | ±0.05 | ±0.05 |
| ZERO (Farina et al., 2024) | × | ±0.15 | ±0.20 | ±0.08 | ±0.06 | ±0.03 | ±0.04 | ±0.05 | ±0.08 | ±0.11 | ±0.03 |
| **SE** | × | ±0.12 | ±0.16 | ±0.16 | ±0.14 | ±0.01 | ±0.03 | ±0.04 | ±0.09 | ±0.02 | ±0.00 |
| TPT (Shu et al., 2022) | ✓ | ±0.00 | ±0.10 | ±0.26 | ±0.20 | ±0.11 | ±0.08 | ±0.09 | ±0.11 | ±0.07 | ±0.01 |
| C-TPT (Yoon et al., 2024) | ✓ | ±0.12 | ±0.06 | ±0.06 | ±0.04 | ±0.03 | ±0.05 | ±0.13 | ±0.01 | ±0.06 | ±0.03 |
| RLCF (Zhao et al., 2024) | ✓ | ±0.30 | ±0.02 | ±0.12 | ±0.11 | ±0.10 | ±0.24 | ±0.08 | ±0.03 | ±0.05 | ±0.03 |
| TTL (Imam et al., 2025) | ✓ | ±0.00 | ±0.08 | ±0.00 | ±0.24 | ±0.05 | ±0.26 | ±0.10 | ±0.21 | ±0.11 | ±0.08 |
| TPS (Sui et al., 2025) | ✓ | ±0.32 | ±0.24 | ±0.08 | ±0.04 | ±0.07 | ±0.23 | ±0.14 | ±0.07 | ±0.02 | ±0.05 |
| R-TPT (Sheng et al., 2025b) | ✓ | ±0.32 | ±0.12 | ±0.14 | ±0.29 | ±0.11 | ±0.24 | ±0.09 | ±0.11 | ±0.06 | ±0.02 |
| STS (Dafnis & Metaxas, 2025) | ✓ | ±0.15 | ±0.08 | ±0.22 | ±0.20 | ±0.05 | ±0.12 | ±0.11 | ±0.14 | ±0.04 | ±0.07 |
| **USE** | ✓ | ±0.09 | ±0.04 | ±0.10 | ±0.04 | ±0.07 | ±0.16 | ±0.02 | ±0.07 | ±0.05 | ±0.02 |

*Table 18.* Standard deviations on fine-grained datasets using the RN-50 backbone.

| Method | Optim. | DTD | Flower102 | Caltech101 | Aircraft | Pets | UCF101 | Cars | EuroSAT | SUN397 | Food101 |
|---|---|---|---|---|---|---|---|---|---|---|---|
| MTA (Zanella & Ben Ayed, 2024b) | × | ±0.03 | ±0.08 | ±0.12 | ±0.03 | ± 0.14 | ±0.33 | ±0.09 | ±0.02 | ±0.08 | ±0.03 |
| ZERO (Farina et al., 2024) | × | ±0.68 | ±0.08 | ±0.18 | ±0.20 | ±0.26 | ±0.03 | ±0.12 | ±0.16 | ±0.05 | ±0.02 |
| **SE** | × | ±0.00 | ±0.06 | ±0.16 | ±0.03 | ±0.03 | ±0.01 | ±0.06 | ±0.04 | ±0.01 | ±0.02 |
| TPT (Shu et al., 2022) | ✓ | ±0.24 | ±0.08 | ±0.22 | ±0.48 | ±0.14 | ±0.11 | ±0.01 | ±0.09 | ±0.04 | ±0.09 |
| C-TPT (Yoon et al., 2024) | ✓ | ±0.03 | ±0.24 | ±0.08 | ±0.29 | ±0.03 | ±0.16 | ±0.02 | ±0.07 | ±0.00 | ±0.02 |
| RLCF (Zhao et al., 2024) | ✓ | ±0.12 | ±0.30 | ±0.08 | ±0.15 | ±0.14 | ±0.01 | ±0.17 | ±0.07 | ±0.01 | ±0.09 |
| TPS (Sui et al., 2025) | ✓ | ±0.12 | ±0.30 | ±0.22 | ±0.09 | ±0.27 | ±0.01 | ±0.09 | ±0.14 | ±0.00 | ±0.01 |
| R-TPT (Sheng et al., 2025b) | ✓ | ±0.00 | ±0.28 | ±0.00 | ±0.03 | ±0.24 | ±0.11 | ±0.02 | ±0.10 | ±0.19 | ±0.02 |
| STS (Dafnis & Metaxas, 2025) | ✓ | ±0.15 | ±0.16 | ±0.10 | ±0.06 | ±0.00 | ±0.25 | ±0.11 | ±0.22 | ±0.18 | ±0.08 |
| **USE** | ✓ | ±0.12 | ±0.04 | ±0.00 | ±0.27 | ±0.04 | ±0.03 | ±0.16 | ±0.03 | ±0.02 | ±0.04 |

# G. Complete Experiment Results

In Section 4, we report summarized results on IN&O, FGVC and Avg. benchmarks, where the performance is presented as the average classification accuracy over ImageNet and its OOD variants, over fine-grained datasets and their average, respectively. In this section, we present the complete experimental results in detail.

All experiments are conducted using the ViT-B/16 backbone. Specifically, we report classification accuracy with CoOp initialization on ImageNet and its OOD variants in Table 19. As well as on fine-grained datasets, which is presented in Table 20.

We further report the complete ablation results on ImageNet and its OOD variants in Table 21, and on fine-grained datasets in Table 22. Finally, we compare classification accuracy with and without the proposed SE strategy on ImageNet and its OOD variants in Table 23, as well as on fine-grained datasets in Table 24.

*Table 19.* Classification Accuracy (%) on ImageNet and OOD variants datasets using CoOp with the ViT-B/16 backbone.

| Method | ImageNet | -A | -V2 | -R | -Sketch |
|---|---|---|---|---|---|
| CoOp (Zhou et al., 2022b) | 71.86 | 50.24 | 64.96 | 75.71 | 48.24 |
| MTA (Zanella & Ben Ayed, 2024b) | 74.17 | 58.71 | 66.85 | 78.81 | 50.25 |
| ZERO (Farina et al., 2024) | 74.31 | 61.55 | 67.06 | 79.02 | 49.97 |
| **SE** | 74.27 | 60.83 | 67.20 | 79.08 | 50.29 |
| TPT (Shu et al., 2022) | 73.79 | 57.42 | 66.60 | 78.51 | 49.75 |
| C-TPT (Yoon et al., 2024) | 73.21 | 53.19 | 66.03 | 77.39 | 49.46 |
| RLCF (Zhao et al., 2024) | 72.43 | 60.57 | 66.25 | 78.81 | 48.95 |
| TTL (Imam et al., 2025) | 74.16 | 59.90 | 67.03 | 79.25 | 50.10 |
| TPS (Sui et al., 2025) | 73.99 | 60.19 | 66.84 | 78.94 | 50.04 |
| R-TPT (Sheng et al., 2025b) | 73.82 | 59.66 | 66.59 | 78.43 | 49.24 |
| STS (Dafnis & Metaxas, 2025) | 73.94 | 63.08 | 66.93 | 78.83 | 49.47 |
| **USE** | 74.15 | 60.26 | 67.16 | 78.93 | 50.27 |

*Table 20.* Classification Accuracy (%) on fine-grained datasets using the CoOp with ViT-B/16 backbone.

| Method | DTD | Flower102 | Caltech101 | Aircraft | Pets | UCF101 | Cars | EuroSAT | SUN397 | Food101 |
|---|---|---|---|---|---|---|---|---|---|---|
| CoOp (Zhou et al., 2022b) | 39.84 | 68.53 | 90.83 | 20.01 | 89.62 | 68.81 | 64.66 | 41.73 | 64.67 | 83.96 |
| MTA (Zanella & Ben Ayed, 2024b) | 40.34 | 69.06 | 92.29 | 20.36 | 90.09 | 69.40 | 67.09 | 42.08 | 66.08 | 84.56 |
| ZERO (Farina et al., 2024) | 40.57 | 68.66 | 92.58 | 19.80 | 89.85 | 68.17 | 66.53 | 42.32 | 65.86 | 83.77 |
| **SE** | 40.78 | 69.33 | 92.41 | 20.13 | 90.16 | 69.31 | 67.11 | 45.90 | 65.97 | 84.56 |
| TPT (Shu et al., 2022) | 40.22 | 69.45 | 92.62 | 20.12 | 89.81 | 69.19 | 67.37 | 48.64 | 65.94 | 84.40 |
| C-TPT (Yoon et al., 2024) | 40.40 | 69.43 | 92.90 | 20.37 | 89.89 | 68.65 | 65.56 | 45.59 | 65.95 | 84.23 |
| RLCF (Zhao et al., 2024) | 44.06 | 68.47 | 93.88 | 22.31 | 89.52 | 67.75 | 67.25 | 44.92 | 67.09 | 84.79 |
| TTL (Imam et al., 2025) | 40.57 | 68.49 | 91.62 | 19.49 | 89.73 | 68.69 | 65.79 | 46.53 | 65.08 | 83.90 |
| TPS (Sui et al., 2025) | 40.43 | 69.14 | 91.95 | 20.30 | 90.08 | 69.09 | 66.94 | 45.27 | 65.67 | 84.52 |
| R-TPT (Sheng et al., 2025b) | 40.16 | 68.82 | 92.64 | 19.62 | 89.94 | 68.69 | 67.26 | 43.56 | 66.00 | 83.94 |
| STS (Dafnis & Metaxas, 2025) | 39.66 | 67.93 | 92.58 | 18.75 | 89.23 | 67.37 | 66.78 | 42.66 | 65.03 | 83.06 |
| **USE** | 40.81 | 69.35 | 92.48 | 20.13 | 90.13 | 69.89 | 67.67 | 47.75 | 66.29 | 84.66 |

*Table 21.* Ablation study on ViT-B/16 evaluating the impact of different optimization and inference components.

| Optim. | Inference | ImageNet | -A | -V2 | -R | -Sketch |
|---|---|---|---|---|---|---|
| - | standard | 66.72 | 47.83 | 60.94 | 73.99 | 46.10 |
| - | uniform | 69.63 | 61.43 | 64.70 | 77.75 | 48.73 |
| - | **SE** | 69.15 | 59.15 | 64.10 | 77.34 | 48.53 |
| MEM | standard | 68.92 | 54.59 | 63.45 | 77.06 | 47.84 |
| MEM | uniform | 69.27 | 61.72 | 64.32 | 77.59 | 48.39 |
| MEM | **SE** | 69.48 | 60.91 | 64.45 | 77.83 | 48.65 |
| RCE | standard | 69.04 | 53.37 | 63.41 | 76.80 | 48.03 |
| RCE | uniform | 69.63 | 61.43 | 64.70 | 77.75 | 48.73 |
| RCE | **SE** | 69.72 | 59.79 | 64.60 | 77.89 | 48.91 |

*Table 22.* Ablation study on ViT-B/16 evaluating the impact of different optimization and inference components.

| Optim. | Inference | DTD | Flower102 | Caltech101 | Aircraft | Pets | UCF101 | Cars | EuroSAT | SUN397 | Food101 |
|---|---|---|---|---|---|---|---|---|---|---|---|
| - | standard | 44.33 | 67.32 | 93.95 | 23.85 | 88.20 | 65.19 | 65.56 | 42.05 | 62.58 | 83.66 |
| - | uniform | 46.04 | 66.22 | 93.85 | 24.84 | 86.89 | 67.13 | 67.42 | 38.62 | 65.04 | 83.44 |
| - | **SE** | 46.45 | 66.91 | 94.36 | 25.13 | 87.86 | 67.75 | 67.86 | 44.44 | 65.02 | 84.44 |
| MEM | standard | 47.05 | 68.64 | 94.06 | 23.30 | 87.30 | 68.38 | 66.59 | 42.90 | 65.51 | 84.64 |
| MEM | uniform | 46.90 | 68.05 | 93.94 | 23.03 | 86.90 | 67.79 | 66.90 | 37.84 | 65.28 | 83.68 |
| MEM | **SE** | 47.16 | 68.31 | 94.12 | 23.70 | 87.09 | 68.12 | 67.42 | 40.09 | 65.61 | 84.33 |
| RCE | standard | 47.49 | 69.27 | 94.30 | 23.96 | 87.94 | 68.28 | 66.55 | 44.97 | 65.55 | 84.69 |
| RCE | uniform | 47.61 | 68.07 | 94.18 | 24.09 | 87.39 | 67.98 | 67.79 | 40.19 | 65.73 | 83.81 |
| RCE | **SE** | 47.61 | 68.78 | 94.50 | 24.35 | 87.94 | 68.68 | 67.72 | 43.31 | 65.89 | 84.71 |

*Table 23.* Classification accuracy (%) on ImageNet and OOD variants with the ViT-B/16 backbone for representative TTA methods, with and without the proposed Self-Ensembling (SE) strategy.

|  | ImageNet | -A | -V2 | -R | -Sketch |
|---|---|---|---|---|---|
| TPT (Shu et al., 2022) | 68.92 | 54.59 | 63.45 | 77.06 | 47.84 |
| SE | 69.48 | 60.91 | 64.45 | 77.83 | 48.65 |
| C-TPT (Yoon et al., 2024) | 68.45 | 51.15 | 62.66 | 75.78 | 47.42 |
| SE | 69.67 | 60.60 | 64.56 | 77.75 | 48.77 |
| R-TPT (Sheng et al., 2025b) | 69.32 | 57.61 | 63.98 | 76.90 | 47.70 |
| SE | 69.16 | 62.01 | 64.28 | 77.60 | 48.27 |
| STS (Dafnis & Metaxas, 2025) | 68.77 | 61.37 | 64.21 | 77.02 | 48.09 |
| SE | 69.16 | 59.80 | 64.22 | 77.33 | 48.59 |

*Table 24.* Classification accuracy (%) on fine-grained datasets with the ViT-B/16 backbone for representative TTA methods, with and without the proposed Self-Ensembling (SE) strategy.

|  | DTD | Flower102 | Caltech101 | Aircraft | Pets | UCF101 | Cars | EuroSAT | SUN397 | Food101 |
|---|---|---|---|---|---|---|---|---|---|---|
| TPT (Shu et al., 2022) | 47.05 | 68.64 | 94.06 | 23.30 | 87.30 | 68.39 | 66.59 | 42.90 | 65.51 | 84.64 |
| SE | 47.16 | 68.31 | 94.12 | 23.70 | 87.09 | 68.12 | 67.42 | 40.09 | 65.61 | 84.33 |
| C-TPT (Yoon et al., 2024) | 45.09 | 69.65 | 93.69 | 24.05 | 88.20 | 65.03 | 65.81 | 42.45 | 64.46 | 83.14 |
| SE | 45.45 | 69.27 | 94.28 | 25.55 | 87.79 | 66.36 | 67.27 | 40.91 | 65.28 | 83.84 |
| R-TPT (Sheng et al., 2025b) | 46.37 | 68.25 | 93.81 | 23.75 | 86.95 | 67.59 | 66.85 | 35.07 | 65.45 | 84.26 |
| SE | 46.87 | 67.95 | 93.85 | 22.76 | 86.59 | 67.50 | 66.58 | 37.87 | 65.01 | 83.58 |
| STS (Dafnis & Metaxas, 2025) | 46.19 | 65.90 | 93.57 | 24.77 | 86.64 | 66.89 | 67.14 | 38.35 | 64.88 | 83.04 |
| SE | 46.66 | 66.83 | 94.16 | 25.17 | 87.64 | 67.72 | 67.98 | 44.20 | 65.09 | 84.27 |

