# OpenReview forum: "USE: A Unified Self-Ensembling Framework for Test-Time Prompt Tuning"
_ICML.cc/2026/Conference — ICML 2026 regular_

### Official Review · Reviewer_XSCC · 2026-03-11

**Soundness:** 3
**Presentation:** 3
**Significance:** 2
**Originality:** 3
**Overall Recommendation:** 3
**Confidence:** 4

**Summary:**

AI models like CLIP sometimes struggle when they encounter images that look different from what they were trained on, this paper asks: what if the model could quickly adapt itself at test time?
The existing approach (TPT) does this by creating lots of augmented versions of the image and making the model agree with itself across all of them. The authors noticed that this process was quietly using the model's own guesses as training targets and those guesses weren't very good, because the original clean image was often being ignored in favor of the augmented copies.
Their fix is simple: give the original image more say. They blend the original image's prediction with the augmented versions, weighting the original more heavily when it seems confident. This produces better training targets and better final predictions.

**Compliance With Llm Reviewing Policy:**

Affirmed.

**Final Justification:**

I've read the additional response carefully, but my assessment remains unchanged. On novelty, the rebuttal restates the contributions rather than providing new evidence of a conceptual departure from TPT. On performance, I note that the authors now frame the goal as exploiting augmentations more fully rather than solving augmentation sensitivity fundamentally, which is actually a more modest claim that aligns with my significance concern rather than resolving it. The paper is a clean and well-executed contribution, but the significance remains limited. Maintaining my score.

**Key Questions For Authors:**

None

**Limitations:**

The whole approach depends heavily on the quality of augmentations. If the augmentations are poor or irrelevant to the domain shift, the pseudo-labels will still be unreliable

**Strengths And Weaknesses:**

Strengths:
1. The proposed method is simple but genuinely effective, improving pseudo-label quality by giving the original image more weight is an intuitive idea that clearly pays off in the results.
2. The paper is well-written and easy to follow. The authors do a good job explaining why their approach works, not just that it works, which is refreshing.

Weaknesses:
The core novelty is relatively incremental. The idea of ensembling weak and strong augmentations and weighting predictions by confidence builds closely on existing work rather than introducing a fundamentally new direction. It's a smart refinement, but unlikely to shift how the field thinks about test-time adaptation.

---

> ### Author Rebuttal · Authors · 2026-03-30
>
> We sincerely appreciate your constructive feedback and your recognition of our approach as “smart” and “genuinely effective.” We have carefully considered each of your comments and provide detailed responses below. Hope these explanations resolve your concerns. Please don't hesitate to let us know if you have any further questions.
>
> ---
> > **Q1**: The core novelty is relatively incremental. The idea of ensembling weak and strong augmentations and weighting predictions by confidence builds closely on existing work rather than introducing a fundamentally new direction. It's a smart refinement, but unlikely to shift how the field thinks about test-time adaptation.
>
> We thank the reviewer for acknowledging our approach as a "smart refinement." While our method utilizes existing concepts like augmentations and confidence weighting, we respectfully argue that our framework introduces fundamental methodological shifts and novel insights to test-time adaptation.
> * **Where to use augmentations.** Existing optimization-based TTA methods often restrict strong augmentations to the optimization stage and discard them during inference. We are the first to shift the focus to leveraging strong augmentations in the final prediction.
> * **How to use augmentations.** While prior methods treat all augmented views equally, our work first reveals the complementary roles of different augmentations. By explicitly prioritizing weak augmentations, we establish an ensembling strategy for more reliable predictions.
>
> Despite its simplicity, our method achieves state-of-the-art performance across various settings. Moreover, we propose a novel unified TTA framework and reinterpret MEM in TPT as a form of implicit self-learning. We believe this simple yet effective design can inspire future research on where and how to use augmentations, as well as advance the current TTA paradigm.
>
> ---
> > **Q2**: The whole approach depends heavily on the quality of augmentations. If the augmentations are poor or irrelevant to the domain shift, the pseudo-labels will still be unreliable.
>
> We agree that the quality of test-time augmentations significantly impacts the performance of existing episodic TTA methods, including our method. Based on the analyses in Fig. 2, we believe our method, especially the self-ensembling strategy, is relatively robust across varying augmentation qualities. Here we further demonstrate this through two distinct scenarios:
>
> - **High-quality strong augmentations**. When strong augmentations yield higher accuracy than the weak augmentation (e.g., on ImageNet-R and DTD), most TTA methods naturally benefit from them. Among these TTA methods, our approach maximizes this utility more effectively. In particular, both SE and USE achieve state-of-the-art results across ViT-B/16 and ResNet-50 architectures on these datasets. Notably, on the DTD dataset, USE yields an accuracy of 47.61% using ViT-B/16, surpassing the second-best method, TPT (47.05%), by a large margin.
> - **Poor-quality strong augmentations.** In challenging cases, where strong augmentations yield lower accuracy than the weak augmentation (e.g., EuroSAT and Food101), existing TTA methods inevitably suffer due to their heavy reliance on these views. Our method mitigates this vulnerability by dynamically leveraging the reliable, complementary signal from the weakly augmented views. As shown in Table 2, on the challenging EuroSAT dataset, several methods degrade sharply below the zero-shot baseline (42.05%), including ZERO (37.19%) and R-TPT (35.07%). In contrast, SE resists this degradation and achieves the best accuracy of 44.44%, outperforming the second-best method (RLCF) by 1.04%. Similarly, on Food101, SE and USE consistently outperform their counterparts, indicating stable performance even when augmentation quality is compromised.
>
> In summary, whether strong augmentations are highly relevant to the domain shift or severely poor, we believe our approach can maintain relatively robust and stable performance. We will include this robustness analysis in the revised manuscript.

---

> > ### Author Rebuttal · Reviewer_XSCC · 2026-04-02
> >
> > Thanks for the rebuttal. I've read it carefully and my position is unchanged.
> > On the novelty concern: I appreciate the authors' argument about "where" and "how" augmentations are used. These are fair points, but they describe design choices rather than a conceptual departure from existing work. The framework is still a refinement of TPT, and I don't think the rebuttal changes that assessment.
> > On the robustness to augmentation quality: the EuroSAT result is striking, but it's doing a lot of work in the argument. Looking across the full tables, the average gains over the best prior methods are generally 0.2–0.6%, which is within or close to the reported standard deviation ranges. On fine-grained with ResNet-50 (Table 4), USE leads TPT by 0.02%. These are not the numbers of a method that has fundamentally solved the augmentation sensitivity problem.
> > Maintaining my score. The paper is a clean, well-executed contribution but the significance remains limited.

---

> > > ### Author Response · Authors · 2026-04-03
> > >
> > > We sincerely thank the reviewer for the prompt feedback and provide our further responses as follows.
> > >
> > > ---
> > > >  Novelty concern.
> > >
> > > 1. To the best of our knowledge, we reinterpret classic marginal entropy minimization from **a novel perspective, implicit self-learning**, which fundamentally differs from all existing TTA methods.
> > > 2. Building upon this self-learning perspective, we develop a simple yet effective self-ensembling strategy that **identifies and prioritizes the role of weak augmentation** during the TTA process.
> > > 3. We identify and exploit an overlooked aspect of existing training-based TTA methods, and apply the self-ensembling strategy again to **enforce consistency between the training and inference phases**.
> > >
> > > Together, these contributions form a unified TTA framework that is neither a collection of mere design choices nor a simple refinement of TPT. Instead, we believe it offers a conceptual departure from existing work and can inspire future research in this direction.
> > >
> > > ---
> > > > Performance gains and robustness to augmentation quality.
> > >
> > > * Our extensive experiments show that USE consistently delivers state-of-the-art or comparable results, achieving the highest overall average accuracy. Specifically, USE outperforms the second-best method, STS, by 0.29% (Table 1) and 0.83% (Table 3), and beats TPT by 0.51% (Table 2) and 0.02% (Table 4). Although USE and TPT share similar average accuracies in Table 4, USE achieves the best performance on 7 individual datasets, whereas TPT leads on only 2 datasets. **Considering the inferior performance of STS and TPT in the other tables, USE clearly demonstrates the most robust performance.**
> > >
> > > * We reinterpret TPT as implicit self-learning and propose a novel, unified TTA framework. **Our approach aims to fully exploit the potential of augmentations for TTA**, rather than to solve the augmentation sensitivity problem fundamentally. Furthermore, empirical results demonstrate that the proposed method is relatively robust to augmentation quality. We thank the reviewer for raising this valuable suggestion and will include such a discussion in the revised manuscript.
> > >
> > > ---
> > > We hope our follow-up responses effectively address your concerns. Should any questions remain, we would be glad to discuss them further.

---

### Official Review · Reviewer_ofb6 · 2026-03-11

**Soundness:** 3
**Presentation:** 3
**Significance:** 3
**Originality:** 3
**Overall Recommendation:** 4
**Confidence:** 3

**Summary:**

This paper revisits Test-Time Prompt Tuning (TPT), a pioneering CLIP-based test-time adaptation method that optimizes textual prompts using multiple test-time augmentations. The authors reveal that TPT can be interpreted as implicitly learning from self-generated pseudo labels. Building on this perspective, they propose a unified self-ensembling framework (USE) that jointly refines both optimization and inference. During optimization, the method introduces a simple self-ensembling strategy that adaptively emphasizes the test image itself over its augmented views to obtain more reliable pseudo labels. The same strategy is further applied at inference time to unify the objectives of both stages. Experiments on multiple datasets show that SE and USE outperform their corresponding counterparts, and SE can also serve as a lightweight training-free TTA method.

**Compliance With Llm Reviewing Policy:**

Affirmed.

**Final Justification:**

I thank the authors for their rebuttal and clarifications. While the rebuttal helped clarify the paper’s contributions, it does not materially change my overall assessment, particularly regarding originality and significance, as I still view the work primarily as a refinement of existing TPT-based ideas rather than a fundamentally new method. Therefore, my final recommendation remains unchanged.

**Key Questions For Authors:**

1. Regarding Eq. (4)-(5), could the authors provide a more rigorous justification for the claim that “the KL term is already optimized and can be dropped”? A clearer justification would help readers better understand this reinterpretation of TPT.
2. The paper fixes the adaptive weighting factor 𝛽 by linearly mapping the relative rank to [0.3,0.7], but the motivation for this specific design is not fully discussed.
3. The paper provides “Analysis of computational efficiency” and states that “SE achieves the lowest memory cost and the fastest inference speed among the compared methods,” but this section is centered on SE, whereas the main framework is USE. Can the authors provide a more systematic efficiency analysis for USE, not only for SE?
4. The ablation suggests that both the optimization design and the inference strategy matter, but it is still not entirely clear whether the main gain comes from better pseudo-labels during optimization, from applying SE at inference time, or from aligning both stages.

**Limitations:**

The method is motivated by emphasizing the weakly augmented view over strong augmentations, but this assumption may not always hold under severe corruption, viewpoint shift, or occlusion. A brief discussion of such failure cases would strengthen the paper.

**Strengths And Weaknesses:**

Strengths
1. The paper “revisits TPT” and provides a new perspective by revealing that its optimization “can be interpreted as implicitly learning from self-generated pseudo labels.”
2. The SE module is versatile and computationally efficient, functioning as a standalone optimization-free TTA method and as a plug-in component for existing baselines, where it yields consistent performance improvements.
3. The paper presents “extensive experiments across multiple datasets” and shows that “SE and USE outperform their counterparts, respectively.”

Weaknesses
1. The paper mainly “revisits TPT” and builds on the view that it “can be interpreted as implicitly learning from self-generated pseudo labels,” so the contribution feels more like a refinement than a fundamentally new method.
2. Algorithm 1 uses 𝐴𝑖(𝑥) without explicitly showing the loop over 𝑖, making the procedure less clear.
3. The current presentation focuses on effectiveness and robustness, but gives little discussion of failure cases or deployment constraints.

---

> ### Author Rebuttal · Authors · 2026-03-30
>
> We sincerely thank the reviewer for the constructive feedback and for recognizing our SE module as 'versatile and computationally efficient' with 'extensive' experiments. We provide detailed responses below and hope they successfully resolve your concerns.
>
> ---
> > **Q1**: The paper mainly revisits TPT... so the contribution feels more like a refinement than a fundamentally new method.
>
> While our method is inspired by revisiting TPT, our contributions go beyond incremental refinement and instead introduce both conceptual and methodological advances：
> * **The significance of revisiting TPT**. We build upon TPT because it has not only inspired a wide range of methods, but also remains highly competitive to date.
> * **Theoretical insights.** Our analysis uncovers the critical, yet previously overlooked, importance of prioritizing the weakly augmented view. Furthermore, we mathematically reinterpret MEM from a self-learning perspective, exposing its implicit pseudo-labeling mechanism. Together, these insights offer promising new directions for future TTA research.
> * **Methodological contributions.** Motivated by these insights, we propose an adaptive SE strategy for more reliable pseudo-label estimation and improved prediction. Moreover, we unify the optimization and inference stages within the USE framework, leading to a novel TTA paradigm.
>
> ---
> > **Q2**: Could the authors discuss failure cases and deployment constraints, particularly when the weakly augmented view is compromised (e.g., severe corruption, ...)?
>
> Thank the reviewer for raising this point. We will include a limitations section in the revision to discuss the following:
> * **Deployment constraints**. Like most optimization-based TTA methods, generating and processing multiple augmented views introduces computational overhead. However, this cost remains marginal relative to the optimization stage itself.
> * **Failure cases**. Our strategy prioritizes the weakly augmented view, and severe corruption to this image can degrade performance. For example, ImageNet-A contains highly compromised examples where strong augmentations actually outperform weak ones (60.16% vs. 47.83%). Consequently, as shown in Tables 1 and 3, USE achieves SOTA on all IN&O datasets except ImageNet-A.
>
> ---
> > **Q3**: Regarding Eq. (4)-(5)... A clearer justification would help readers better understand this reinterpretation of TPT.
>
> We introduce an auxiliary distribution $q$ to decompose the MEM into $RCE(p;q) + KL(p || q)$. By specifically defining $q$ as the stop-gradient version of the current prediction $p$, the two distributions become identical. Consequently, the KL divergence **achieve optimal**, making MEM approximately equivalent to RCE optimization. This approximation is further **empirically validated** in the Appendix, where RCE is shown to be effectively equivalent to MEM.
>
> ---
> > **Q4**: The paper fixes the adaptive weighting factor $\beta$ by linearly mapping the relative rank to [0.3,0.7], but the motivation for this specific design is not fully discussed.
>
> Thank the reviewer for raising this point. The detailed rationale behind this formulation is also discussed in our response to Reviewer `ePCq` (Q4). Specifically, applying a scaling factor of $\gamma=0.4$ maps the adaptive weights to the interval [0.3, 0.7], consistently providing superior results across different settings. Furthermore, our method **is not sensitive** to this configuration. When $\gamma=0.2$ ([0.4,0.6]) or  $\gamma=0.6$ ([0.2,0.8]) is applied, the average accuracy across IN&O and FGVC remains 64.77% and 64.72% respectively, demonstrating a clear advantage over the baselines.
>
> ---
> > **Q5**: While the efficiency of SE is established, can a more systematic analysis of computational efficiency be provided for USE?
>
> We appreciate the reviewer's suggestion and will include this re-evaluated analysis in the revision.
>
> Table. Computational efficiency comparison.
> | Method | Time [s] | Mem. [GB] | Acc. [%] | Gains |
> | :--- | :---: | :---: | :---: | :---: |
> | CLIP | 0.015 | 0.853 | 63.67| - |
> | TPT |0.200| 6.579 | 64.84 | 1.17 |
> |  R-TPT |0.298 |6.579 |63.83| 0.16|
> | C-TPT | 0.202 | 6.579 | 64.16 | 0.49 |
> | RLCF  |2.697 | 7.329 | 64.46 | 0.79|
> | TTL  | 0.180 | **6.473** | 63.93| 0.26 |
> | USE | **0.144** | 6.568| **65.35** | **1.68** |
>
> ---
> > **Q6**: The ablation... whether the main gain comes from better pseudo-labels during optimization, from applying SE at inference time, or from aligning both stages.
>
> Thank you for the good question. As shown in Table 6, while all three components contribute, the main gain stems from **applying SE at inference time**, which improves the RCE from 63.71% to 64.77%. Furthermore, RCE+SE(64.77%) outperforms MEM+SE(64.43%) by preventing the performance degradation that occurs when SE is applied directly to MEM. Finally, unifying the two stages improves performance from RCE+uniform(64.56%) to RCE+SE(64.77%), leading to the best overall results. We will clarify this in the revision.

---

> > ### Author Rebuttal · Reviewer_ofb6 · 2026-04-02
> >
> > My main concerns have been adequately addressed, and the authors have provided clear explanations.

---

> > > ### Author Response · Authors · 2026-04-02
> > >
> > > We are pleased that our responses have adequately addressed your concerns, and we are truly grateful for your continued support. If you feel it is appropriate, we would greatly appreciate it if you could raise your score. Your support remains invaluable to us.

---

### Official Review · Reviewer_ePCq · 2026-03-12

**Soundness:** 3
**Presentation:** 3
**Significance:** 2
**Originality:** 2
**Overall Recommendation:** 4
**Confidence:** 3

**Summary:**

This paper investigates test-time prompt tuning (TPT) and show that its optimization process can be interpreted as a form of self-refinement using pseudo-labels derived from the model’s own predictions. Based on this insight, they propose Unified Self-Ensembling (USE), a framework that enhances both optimization and inference during test-time adaptation. The method aggregates predictions from augmented views while emphasizing the original image to generate more reliable pseudo-labels. Extensive experiments demonstrate that USE consistently improves the robustness and performance of vision–language models across multiple tasks and distribution shift scenarios.

**Compliance With Llm Reviewing Policy:**

Affirmed.

**Final Justification:**

The paper is well-written, and the figures and tables are clear. The authors' rebuttal effectively addressed my previous concerns, especially in the second round, which resolved most of my issues to a large extent.

Regarding the previously noted concern about limited novelty, I acknowledge that the proposed framework is the first to reinterpret the classic marginal entropy minimization from a self-learning perspective. Given its simple and direct design, it may still provide additional insights to the field. Furthermore, the authors' willingness to revise the potentially confusing term "Unified" is appreciated. Based on these considerations, I have increased my score.

**Key Questions For Authors:**

Please refer to the Weakness section above. If the authors can address these concerns, I would consider raising the rating.

**Limitations:**

yes

**Strengths And Weaknesses:**

## Strengths
1. The paper is clearly structured, making the methodology and motivation clear and accessible.
2. The discussion on how to obtain more reliable pseudo-labels is thoughtful and well-motivated.
3. The method is evaluated on multiple benchmarks(e.g., ImageNet-A, ImageNet-V2, etc.) and consistently outperforms strong baselines.
## Weaknesses
1. The proposed self-ensembling strategy appears relatively simple, and weighting prediction results is not particularly novel, which may limit the overall novelty of the method.
2. In Section 3.3, the authors explain that the term *“Unified”* refers to the fact that *“USE unifies the training and inference objectives within a SE framework.”* However, maintaining consistent model behavior between training and inference is a common strategy for improving test-time adaptation (TTA) performance. The term *“Unified”* may therefore feel somewhat unusual in this context, as it is more commonly used to describe a framework that unifies multiple tasks or problem settings.
3. The bar chart in Figure 3 does not include numerical values, which makes it difficult to clearly compare the performance differences across different thresholds. In addition, the concept of *“adaptive”* should be explicitly discussed when analyzing this figure to avoid potential misunderstandings for readers.
4. In Equation (7), the parameter $\beta$ is defined as $\beta = 0.5 + 0.4 × (\delta − 0.5)$. It would be helpful if the authors could clarify how this formulation was determined and share the empirical considerations or experience behind this design choice.

---

> ### Author Rebuttal · Authors · 2026-03-30
>
> Thank you for the helpful feedback and thoughtful guidance. We feel especially encouraged by your recognition of our presentation and experiments. We hope our detailed responses successfully resolve your concerns. If there are any remaining uncertainties, we would be happy to clarify them.
>
> ---
> > **Q1**: The proposed self-ensembling strategy appears relatively simple, and weighting prediction results is not particularly novel, which may limit the overall novelty of the method.
>
> While the weighting mechanism itself appears simple, we respectfully emphasize that this strategy introduces two fundamentally new perspectives to identify and utilize augmentations.
> * New perspective on **where to use augmentations.** While previous methods primarily focus on using augmentations during the optimization, we are the first to shift the focus to their critical role during inference, explicitly leveraging self-ensembling for the final prediction.
> * New identification on **how to use augmentations**. We are the first to uncover the complementary relationship between weak and strong augmentations. By explicitly prioritizing the weak augmentation, the SE strategy achieves highly stable and superior performance.
>
> Moreover, we propose a new framework that unifies the optimization and inference stages under the same objective, and reinterpret MEM as implicit self-learning, offering valuable perspectives on TTA problems.
>
> ---
> > **Q2**: Maintaining consistent model behavior between training and inference is a common strategy for improving test-time adaptation (TTA) performance. The term “Unified” may therefore feel somewhat unusual in this context, as it is more commonly used to describe a framework that unifies multiple tasks or problem settings.
>
> We sincerely thank the reviewer for this insightful consideration. We agree that aligning training and inference is a desirable strategy for ML problems. However, we respectfully highlight that **it remains underexplored** in episodic TTA for VLMs, where existing methods typically optimize on augmented views but perform inference solely on the test image.
> We understand that the term "Unified" is often associated with multiple tasks or problems. However, we used it to **specifically emphasize the novel alignment** of the optimization stage and inference stage in our work. To avoid potential ambiguity, we will add a clarification in the revised manuscript.
>
> ---
> > **Q3**: The bar chart in Figure 3 does not include numerical values, and the concept of “adaptive” should be explicitly discussed when analyzing this figure.
>
> We thank the reviewer for the reminder. We will update Figure 3 in the revision. Specifically, "adaptive" represents the SE strategy that dynamically adjusts ensembling weights.
>
> ---
> > **Q4**: In Equation (7), the parameter $\beta$ is defined as $\beta = 0.5+ 0.4 \cdot (\delta-0.5)$ . It would be helpful if the authors could clarify how this formulation was determined and share the empirical considerations or experience behind this design choice.
>
> We thank the reviewer for raising this important point. While Equation (7) is introduced as a linear mapping in the main paper for brevity, we detail the underlying motivation and empirical considerations below:
> * **Formulation Motivation.** As shown in Figure 2, although augmentation effectiveness varies across datasets, weak and strong views consistently provide complementary information. A simple average yields superior results but lacks the flexibility to reweight dynamically. To address this, we design an **adaptive reweighting strategy that adjusts the coefficients around the $0.5$ based on the quality of the weak augmentation**. Specifically, it is formulated as $\beta = 0.5 + \gamma \cdot (\delta - 0.5)$. The scaling factor $\gamma$ controls the dynamic width. Fixing $\gamma = 0.4$, we constrains the weight within $[0.3, 0.7]$. This prevents weight collapse (i.e., $0$ or $1$), ensuring that neither view is entirely discarded.
> * **Empirical Consideration.** We also explore **more complex variants**, such as square root form ($\beta = 0.5 \pm 0.4 \cdot \sqrt{|\delta - 0.5|}$) and squared form ($\beta = 0.5 \pm 0.4 \cdot (\delta - 0.5)^2$). Experiments show that these more complex versions may provide performance gains, but we adopt the linear design as a simple and effective choice.
>
> Table. Classification accuracy (%) of SE variants on different datasets using the ViT-B/16 backbone.
> | Method | IN&O | FGVC | Avg. |
> | :--- | :--- | :--- | :--- |
> | SE-Square Root |  63.38|  65.01| 64.19 |
> | SE-Squared | **63.81** | 64.94 | **64.38** |
> | **SE** | 63.65 | **65.02** | 64.34 |

---

> > ### Author Rebuttal · Reviewer_ePCq · 2026-04-03
> >
> > Thanks for your rebuttal. Regarding the response to W1, I have carefully examined the design choices and motivation of USE. As also noted by **Reviewer XSCC**, the novelty of USE still appears somewhat limited.
> >
> > In addition, I remain unconvinced that the term "Unified" is an appropriate description in this context. I do not agree that the setting of "optimize on augmented views but perform inference solely on the test image" leads to a meaningful inconsistency between the optimization and inference stages. On the contrary, such a design is common and can effectively evaluate model robustness and generalization. If the authors intend to emphasize this subtle distinction, a term such as "consistent" may be more precise and less potentially misleading than "unified."
> >
> > Overall, the paper is clearly written and the experimental evaluation is fairly comprehensive. However, the above concerns still leave me somewhat hesitant, particularly regarding the use of the term "Unified." If the authors can address these issues, I would consider raising my rating.

---

> > > ### Author Response · Authors · 2026-04-03
> > >
> > > Thank you for your constructive feedback. We deeply appreciate your time and the opportunity to further clarify our contributions. We provide our point-to-point responses below:
> > >
> > > ---
> > > > The novelty of USE still appears somewhat limited.
> > >
> > > To the best of our knowledge, we are the first to **reinterpret the classic marginal entropy minimization from a self-learning perspective**, revealing its nature of learning towards a self-generated pseudo-label. Building upon this key insight, we identify the crucial role of weak augmentation and propose **a simple yet effective self-ensembling strategy** to generate more reliable pseudo-labels. By maintaining the optimization and inference objectives in the same form, we establish **a consistent TTA framework**. We respectfully argue that this novel self-learning perspective, combined with the practical effectiveness of the self-ensembling strategy and consistent self-ensembling framework, constitutes a solid contribution to the field.
> > >
> > > ---
> > > > The term "Unified" is an appropriate description.
> > >
> > > We agree that using the term "Unified" may indeed lead to potential misunderstandings, while "Consistent" is a more precise term to use. Accordingly, we will carefully revise our manuscript to replace "Unified" with "Consistent" to prevent any future confusion.
> > >
> > > ---
> > > Thank you again for your valuable feedback. We hope our follow-up response can address your remaining concerns, and we would greatly appreciate it if you could reconsider your score.

---

### Official Review · Reviewer_XWbU · 2026-03-23

**Soundness:** 3
**Presentation:** 3
**Significance:** 3
**Originality:** 3
**Overall Recommendation:** 4
**Confidence:** 4

**Summary:**

The paper introduces USE (Unified Self-Ensembling), a framework for CLIP-based test-time adaptation. It reinterprets prompt tuning as self-training and proposes an entropy-based self-ensembling (SE) strategy that adaptively weights weak and strong augmentations. USE unifies optimization and inference, achieving state-of-the-art accuracy across ImageNet OOD and fine-grained datasets with high efficiency.

**Compliance With Llm Reviewing Policy:**

Affirmed.

**Key Questions For Authors:**

Please see the Weakness part.

**Limitations:**

The authors fail to formally discuss the limitations of their approach, such as, the additional inference overhead (e.g., 64 forward passes per image) compared to standard zero-shot inference.

**Strengths And Weaknesses:**

**Strengths**

- Simple and Effective Pseudo-Label Refinement: The proposed SE strategy provides an intuitive way to improve pseudo-label reliability by prioritizing the original image (weak augmentation) based on its prediction entropy.

- Comprehensive Evaluation and Ablation: The work is backed by extensive experiments across ImageNet OOD variants and 10 fine-grained datasets, demonstrating consistent state-of-the-art performance and verifying the contribution of each component.

- Efficiency via Skip Mechanism: The framework introduces a smart "skip technique" that bypasses optimization when initial predictions are deemed reliable. This reduces computational overhead by over 60% without sacrificing accuracy, making it highly practical for real-world deployment.


**Weaknesses**

- Marginal Improvement of USE over SE: While USE outperforms SE, the margin is relatively small in several scenarios.


- Hyperparameter Sensitivity: The model exhibits sensitivity to the scaling factor $\gamma$, which affects the ensembling weight $\beta$. As shown in Fig 6, while $\gamma = 0.4$ maintains stable performance on fine-grained (FGVC) datasets, increasing it to $\gamma = 1.0$ leads to performance degradation on ImageNet OOD variants. This indicates that the optimal parameter configuration may vary significantly across different domains, challenging the framework's "plug-and-play" robustness.

---

> ### Author Rebuttal · Authors · 2026-03-30
>
> Thank you for your constructive review. We appreciate your recognition of the simplicity and effectiveness of our SE strategy, as well as our comprehensive experiments. Below, we address your specific concerns and hope our responses clarify any questions. Please feel free to let us know if anything remains unclear.
>
> ---
> > **Q1**: While USE outperforms SE, the margin is relatively small in several scenarios.
>
> We appreciate your careful observation. While the numerical margin may appear modest in certain scenarios, we respectfully highlight that USE provides stable empirical improvements and offers insightful value.
> * **Consistent empirical gains.** As shown in Tables 1–4, USE provides consistent improvements. **USE yields substantial gains in most settings.** On IN&O datasets, USE improves average accuracy from 63.65% to 64.18% (ViT-B/16, Table 1) and from 47.80% to 48.41% (ResNet-50, Table 3). On the FGVC datasets, accuracy increases from 56.99% to 57.82% (ResNet-50, Table 4). The smaller gain observed on FGVC with ViT-B/16 (65.02% to 65.35%, Table 2) occurs in a setting with inherently tight margins, where the gap between the second-best TPT and third-best MTA is only 0.07%, making our improvement of 0.33% between SE and USE competitive.
> * **New insights.** Beyond empirical performance, USE reinterprets the classic MEM objective in TPT as implicit self-learning and incorporates SE into a novel unified framework, which we believe may inspire future research in test-time adaptation.
>
> ---
> > **Q2**: The model exhibits sensitivity to the scaling factor $\gamma$, which affects the ensembling weight... This indicates that the optimal parameter configuration may vary significantly across different domains, challenging the framework's "plug-and-play" robustness.
>
> - We appreciate the reviewer's careful analysis. We agree that the optimal $\gamma$ varies across domains, and increasing $\gamma$ leads to performance degradation on the IN&O datasets. However, **empirical results confirm that fixing $\gamma=0.4$ yields highly stable performance.** As Tables 1–4 demonstrate, fixing $\gamma=0.4$ enables both SE and USE to achieve state-of-the-art performance across multiple architectures and benchmarks without requiring any domain-specific tuning.  Specifically, we note that even under our **worst-case configuration ($\gamma=1$)**, USE still achieves an average accuracy of 64.64% across IN&V and FGVC using ViT-B/16. This outperforms the standard baseline of 63.15% ($\beta=1$) by a clear margin.
>
> - Regarding the 'plug-and-play' robustness of the SE strategy, in addition to Table 7, we further provide a deeper analysis of incorporating SE with other TTA methods using different $\gamma$ values below. The `None' column means the results without the SE strategy. Notably, we consistently achieve clear performance gains when incorporating SE, regardless of the value of
> $\gamma$. Also, $\gamma=0.4$ still achieves competitive results, even though it may not be the optimal parameter. We will clarify this robustness in the revised manuscript.
>
> Table. Classification accuracy (%) of different $\gamma$ on the IN&O dataset using the ViT-B/16 backbone.
> | Method | None | 0.2 | 0.4 | 0.6 | 0.8 | 1.0 |
> | :--- | :---: | :---: | :---: | :---: | :---: | :---: |
> | TPT+SE| 62.37 | 64.23 |  64.26 | **64.28** | 63.93| 63.85|
> | C-TPT+SE | 61.09| **64.30** |64.27 | 64.15 | 63.85| 63.72|
>
> ---
> > **Q3**: This paper fails to formally discuss the limitations of their approach, such as the additional inference overhead (e.g., 64 forward passes per image) compared to standard zero-shot inference.
>
> We agree that a formal discussion of limitations is essential, and we thank the reviewer for pointing this out. We acknowledge that the computational cost of 64 forward passes is a limitation of our approach, as it is for existing episodic TTA methods when compared to zero-shot inference. However, **empirical results show that SE maintains competitive performance even when the augmentation budget is reduced.** As reported below, SE with 16 augmentations maintains a competitive 1.09% gain. These results are re-evaluated in a standardized environment. We will add a "Limitations" section in the revised manuscript to formally discuss.
>
> Table. Computational efficiency comparison of different versions of SE on FGVC datasets.
> | Method | Time [s] | Mem. [GB] | Acc. [%] | Gains |
> | :--- | :---: | :---: | :---: |:---: |
> | CLIP| 0.015 | 0.853|  63.67 |   -  |
> | SE-8 | 0.018 | 0.856 | 64.48  |   0.81  |
> | SE-16 | 0.022 | 0.851 | 64.76  |  1.09   |
> | SE-32 | 0.033 | 0.893 | 64.94  |  1.27   |
> | SE-64 | 0.059 | 1.060 | 65.02 |  1.35   |

---

> > ### Author Rebuttal · Reviewer_XWbU · 2026-04-03
> >
> > Thank you for the detailed rebuttal and the additional experiments on robustness and efficiency. I appreciate the stable performance of $\gamma=0.4$ and the commitment to adding a "Limitations" section.
> >
> > However, the lack of a fundamental shift from TPT remains a shared concern among reviewers, as the current marginal improvements over the training-free SE baseline raise questions about the practical necessity of the full USE framework. Therefore, I decide to maintain my current score.

---

> > > ### Author Response · Authors · 2026-04-03
> > >
> > > We sincerely appreciate your recognition of our efforts and thank you for your supportive feedback. Our follow-up responses are provided below:
> > >
> > > > Fundamental shift from TPT.
> > >
> > > * **Conceptual Shift**. To the best of our knowledge, we are the first to reinterpret the classic marginal entropy minimization (MEM) through implicit self-learning. Moreover, we identify the critical role of weak augmentation and leverage it during both the training and inference phases.
> > > * **Methodological Shift.** We reformulate the MEM objective into the reverse cross-entropy (RCE) objective. Besides, we introduce a dynamic self-ensembling strategy that adaptively ensembles different augmentations to achieve superior predictions. Additionally, we ensure consistency between the optimization and inference phases.
> > > Together, these contributions form a consistent TTA framework, which brings a fundamental shift to TPT and can inspire future research in this direction.
> > >
> > > ---
> > > > Practical necessity of the full USE framework.
> > >
> > > We agree that SE is simple and effective. As explained in the manuscript and following responses, USE always provides the best overall performance and offers new insights, such as a novel optimization objective and a consistent TTA framework.
> > >
> > > ---
> > > We thank the reviewer again for your valuable comments, and we will carefully incorporate all your suggestions into the revised manuscript.

---

### Decision · Program_Chairs · 2026-04-30

**Decision:**

Accept (regular)

**Comment:**

The proposed SE scheme provides an intuitive way to improve pseudo-label reliability by prioritising the original image (weak augmentation) based on the prediction entropy. The approach is evaluated on multiple benchmarks (e.g., ImageNet-A, ImageNet-V2, etc.) and consistently outperforms strong baselines.  The method is simple yet genuinely effective, improving pseudo-label quality by giving the original image more weight is an intuitive idea which clearly pays off in the results. Thus most of the reviewers recommend acceptance.